**Subject Area:**
biochemistry/microbiology/molecular biology/ developmental biology

cyst, topoisomerase IIIβ, transcription, *Giardia*, differentiation, DNA-binding protein

**Author for correspondence:**
Chin-Hung Sun
e-mail: chinhsun@ntu.edu.tw

†These authors contributed equally to this study.

# DNA topoisomerase IIIβ promotes cyst generation by inducing cyst wall protein gene expression in *Giardia lamblia*

Chin-Hung Sun[1], Shih-Che Weng[1,†], Jui-Hsuan Wu[1,†], Szu-Yu Tung[1], Li-Hsin Su[1], Meng-Hsuan Lin[1] and Gilbert Aaron Lee[2]

[1]Department of Tropical Medicine and Parasitology, College of Medicine, National Taiwan University, Taipei 100, Taiwan, Republic of China
[2]Department of Medical Research, Taipei Medical University Hospital, Taipei, Taiwan, Republic of China

C-HS, 0000-0002-8604-8085

*Giardia lamblia* causes waterborne diarrhoea by transmission of infective cysts. Three cyst wall proteins are highly expressed in a concerted manner during encystation of trophozoites into cysts. However, their gene regulatory mechanism is still largely unknown. DNA topoisomerases control topological homeostasis of genomic DNA during replication, transcription and chromosome segregation. They are involved in a variety of cellular processes including cell cycle, cell proliferation and differentiation, so they may be valuable drug targets. *Giardia lamblia* possesses a type IA DNA topoisomerase (TOP3β) with similarity to the mammalian topoisomerase IIIβ. We found that TOP3β was upregulated during encystation and it possessed DNA-binding and cleavage activity. TOP3β can bind to the *cwp* promoters *in vivo* using norfloxacin-mediated topoisomerase immunoprecipitation assays. We also found TOP3β can interact with MYB2, a transcription factor involved in the coordinate expression of *cwp1-3* genes during encystation. Interestingly, overexpression of TOP3β increased expression of *cwp1-3* and *myb2* genes and cyst formation. Microarray analysis confirmed upregulation of *cwp1-3* and *myb2* genes by TOP3β. Mutation of the catalytically important Tyr residue, deletion of C-terminal zinc ribbon domain or further deletion of partial catalytic core domain reduced the levels of cleavage activity, *cwp1-3* and *myb2* gene expression, and cyst formation. Interestingly, some of these mutant proteins were mis-localized to cytoplasm. Using a CRISPR/Cas9 system for targeted disruption of *top3β* gene, we found a significant decrease in *cwp1-3* and *myb2* gene expression and cyst number. Our results suggest that TOP3β may be functionally conserved, and involved in inducing *Giardia* cyst formation.

## 1. Introduction

*Giardia lamblia* is a frequent cause of waterborne diarrhoeal diseases in developing countries and in tourists [1,2]. After acute giardiasis, a higher risk of post-infectious irritable bowel syndrome has been reported [3]. Children with chronic giardiasis are vulnerable to malnutrition due to malabsorption, resulting in delayed growth and mental development [4]. A parasitic trophozoite is capable of transforming into a dormant cyst form, in which the cyst wall is essential for transmission of giardiasis during survival in fresh water or the new host's stomach [1].

The small genome suggests *Giardia* as a simplified life form of evolutionary interest [5]. It contains most pathways for life events but with fewer conserved components as compared with yeast [5]. *Giardia* is also a good model for studying single-cell differentiation due to its easy transition between the trophozoite and cyst forms *in vitro* [1,2]. After sensing encystation stimuli, trophozoites

perform a coordinated synthesis of the three cyst wall proteins (CWPs) which are transported through encystation secretory vesicles (ESVs) to form a protective cyst wall [1,2]. Signalling molecules and transcription factors, including CDK2, MYB2 (Myb1-like protein in the *Giardia* genome database), WRKY, PAX1 and E2F1, may play a role in inducing the *cwp* gene expression [6–10]. We also found that a myeloid leukaemia factor (MLF) protein plays an important role in inducing *Giardia* differentiation into cysts [11]. We used our newly developed CRISPR/Cas9 system in *G. lamblia* for targeted disruption of *mlf* gene expression to analyse MLF [11].

Topoisomerases are essential enzymes that can overcome the topological problems of chromosomes during DNA replication, transcription, recombination and mitosis [12,13]. They are involved in cell growth, tissue development and cell differentiation [12–14]. The type I topoisomerases function by cutting one strand of DNA, but type II topoisomerases cut two strands of DNA [12,13]. Therefore, the type I topoisomerases have a weaker relaxation effect than type II [15]. Human topoisomerases IIIα (TOP3α) and IIIβ (TOP3β) belong to the type IA family [16]. The human type IA topoisomerases are monomeric and ATP independent [16]. They create a transient single-stranded DNA break by transesterification of a catalytic Tyr of the cleavage domain and a phosphodiester bond of DNA, and form a covalent 5′ phosphotyrosyl complex with DNA [11,12]. They further act by passing a single strand of DNA through the break to disentangle DNA [11,12]. They prefer to relax negative supercoiled DNA [16]. The N-terminal Toprim domain of bacterial type IA topoisomerases forms active-site region with domain 3, which contains catalytic Tyr residue [17]. The C-terminal zinc ribbon domain of bacterial type IA topoisomerases binds to DNA and interacts with other proteins to unwind DNA [18]. Disruption of yeast topoisomerase III resulted in a significant growth defect [19]. Topoisomerase IIIβ null mutant mice had a shorter lifespan and spleen hypertrophy [20,21]. Disruption of topoisomerase III gene from zebra fish can affect T-cell differentiation [22]. Human type IA topoisomerases are not drug targets, but all other human topoisomerases are important targets for cancer chemotherapy [23]. Many anti-cancer compounds act through inhibiting topoisomerase activity in cancer cells [24]. Many antibiotics can inhibit type II topoisomerase by stabilizing covalent topoisomerase–DNA cleavage complexes, including norfloxacin [25,26].

During *Giardia* encystation, a trophozoite with two nuclei (4N) may differentiate into a cyst with four nuclei (16N) by DNA replication and homologous recombination may occur in the cyst nuclei [1,27]. Because type I topoisomerases play a critical role in cell differentiation [20–22], we asked whether type I topoisomerases could be important for *Giardia* encystation. In our previous study, we found that a *Giardia* type II topoisomerase (TOPO II) is an important factor involved in inducing encystation and the TOPO II inhibitor, etoposide, can inhibit *Giardia* growth and encystation [28]. In this study, we further tried to understand the role of a type IA topoisomerase, TOP3β. We found that overexpression of TOP3β increased expression of *cwp1-3* and *myb2* and cyst formation. Using mutation analysis and a CRISPR/Cas9 system, we found evidence of TOP3β in inducing *Giardia* encystation. We also tested the effect of a type IA topoisomerase inhibitor, norfloxacin, and found that it inhibited *Giardia* growth and cyst formation, and increased the formation of cleavage

complex of TOP3β and DNA. Our results provide insights into the role of TOP3β in activation of *cwp* genes during *Giardia* encystation and into the effect of the TOP3β inhibitor, norfloxacin, on *Giardia* cyst formation and growth.

# 2. Results

## 2.1. Identification and characterization of *top3β* gene

Four putative homologues for topoisomerases have been found in the *G. lamblia* genome database [28]. One is Topo II topoisomerase (open reading frame, orf, 16975). Orfs 15190 and 7615 are annotated as topoisomerase III, which belongs to type IA topoisomerases. Sequence analysis suggests that orfs 15190 and 7615 are similar to human TOP3β and TOP3α, respectively (see below). The last putative topoisomerase homologue is annotated as spo11 type II DNA topoisomerase VI subunit A. We focused on understanding the role of orf 15190 (topoisomerase IIIβ, TOP3β) in *Giardia*. The deduced *Giardia* TOP3β protein contains 973 amino acids with a predicted molecular mass of approximately 107.06 kDa and a pI of 8.40. It has a Toprim domain (residues 2–145) and a DNA topoisomerase domain (residues 159–610) as predicted by Pfam (figure 1*a*) (http://pfam.sanger.ac.uk/) [29]. A Toprim domain is a conserved active region typically found in type IA and type II topoisomerases [17]. A zinc ribbon domain is present in the C terminus of *Giardia* TOP3β (residues 645–973) (electronic supplementary material, figure S1). The C-terminal zinc ribbon domains of bacterial type IA topoisomerases are important for DNA binding and interaction of RNA polymerase [18,30,31]. *Giardia* TOP3β also has a conserved Tyr (residue 328), corresponding to the catalytically important Tyrosines of *Escherichia coli* topoisomerase I (residue 319) and human TOP3β (residue 336) (figure 1*a*; electronic supplementary material, figure S1) [20,32]. *Escherichia coli* topoisomerase III has a unique insertion which is a decatenation loop and not found in *Giardia* and other eukaryotic TOP3 (electronic supplementary material, figure S1) [33]. The full length of *Giardia* TOP3β has 28.73% identity and 41.32% similarity to that of human TOP3β (calculated from electronic supplementary material, figure S1). A phylogenic tree obtained from the alignment of the topoisomerase III proteins from various organisms revealed that *Giardia* TOP3β (15190) is similar to TOP3β from other organisms, and that *Giardia* TOP3α (7615) is similar to TOP3α from other organisms (electronic supplementary material, figure S2).

## 2.2. Encystation-induced expression of the *top3β* gene and perinuclear localization of the TOP3β protein

RT-PCR and quantitative real-time PCR analysis showed that the *top3β* mRNA increased by approximately 1.75-fold in 24 h encysting cells (figure 1*b*). Western blot analysis with anti-TOP3β antibody revealed that the TOP3β level significantly increased during encystation (figure 1*c*).

To determine the expression of TOP3β protein, we prepared construct pPTOP3β, in which the *top3β* gene is controlled by its own promoter and contains an HA epitope tag (approx. 1 kDa) at its C terminus (figure 1*d*), and stably transfected it into *Giardia*. Similar to the expression pattern of the endogenous TOP3β protein (figure 1*c*), the level of

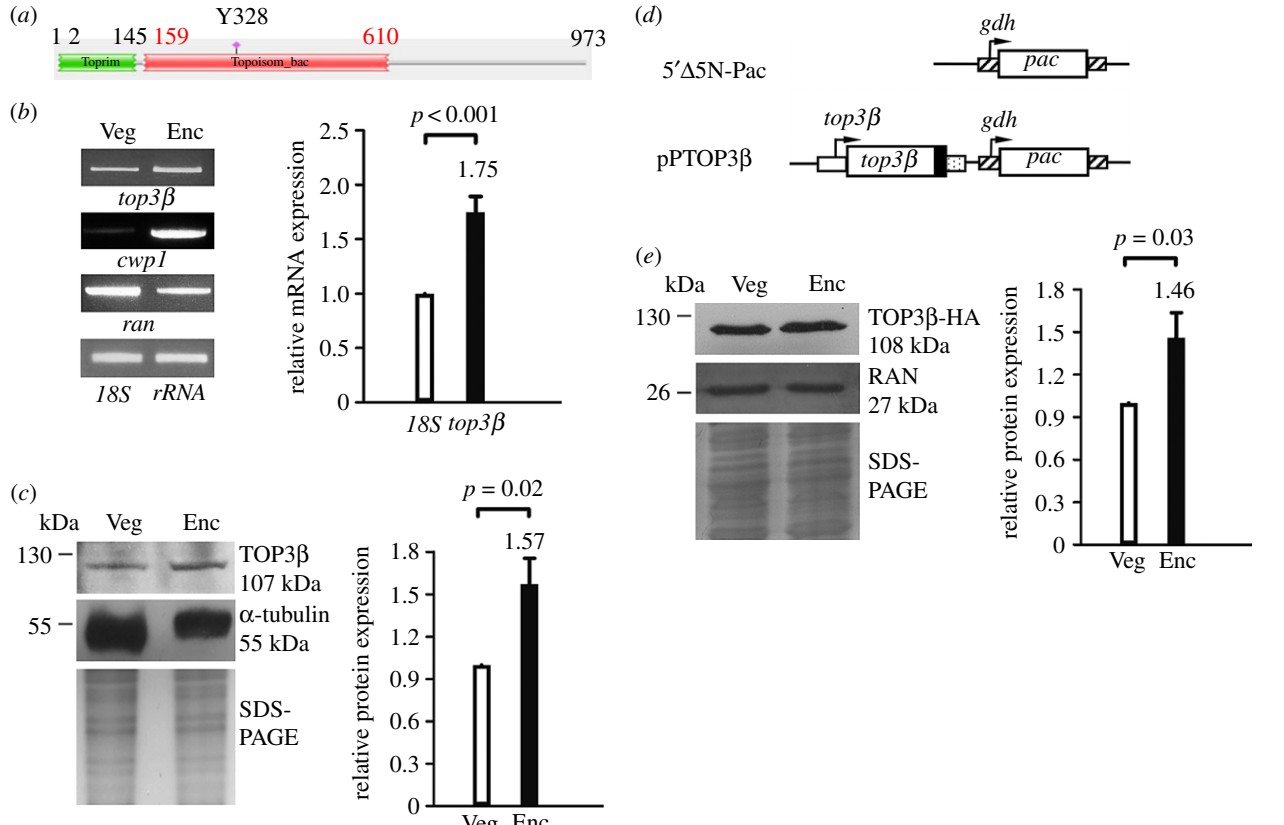

**Figure 1.** Analysis of *top3β* gene expression. (*a*) Schematic of the *Giardia* TOP3β protein. The green box and red box indicate the Toprim domain and Topoisom_bac domain, respectively, as predicted by pfam. The conserved Tyr 328 (Y328) is indicated. (*b*) RT-PCR and quantitative real-time PCR analysis of *top3β* gene expression. RNA samples were prepared from *G. lamblia* wild-type non-transfected WB cells cultured in growth (Veg, vegetative growth) or encystation medium and harvested at 24 h (Enc, encystation). RT-PCR was performed using primers specific for *top3β*, *cwp1*, *ran* and 18S ribosomal RNA (18S rRNA) genes, respectively (left panel). Real-time PCR was performed using primers specific for *top3β* and 18S ribosomal RNA genes, respectively (right panel). Transcript levels were normalized to 18S ribosomal RNA levels. Fold changes in mRNA expression are shown as the ratio of transcript levels in encysting cells relative to vegetative cells. Results are expressed as the means ± 95% confidence intervals (error bars) of at least three separate experiments. $p < 0.05$ was considered significant and the value was shown. As controls, we found that the mRNA expression of *cwp1* and *ran* significantly increased and decreased during encystation, respectively. (*c*) TOP3β level increased during encystation. The wild-type non-transfected WB cells were cultured in growth (Veg, vegetative growth) or encystation medium for 24 h (Enc, encystation) and then subjected to SDS-PAGE and Western blot analysis. The blot was probed with anti-TOP3β and anti-α-tubulin antibodies, respectively. Equal amounts of protein loading were confirmed by SDS-PAGE and Coomassie Blue staining. The α-tubulin level slightly decreased during encystation. The intensity of bands from three Western blot assays was quantified using ImageJ. The ratio of TOP3β protein over the loading control (Coomassie Blue-stained proteins) is calculated. Fold change is calculated as the ratio of the difference between the Enc sample and Veg sample, to which a value of 1 was assigned. Results are expressed as mean ± 95% confidence intervals. $p < 0.05$ was considered significant and the value was shown. (*d*) Diagrams of the 5'Δ5N-Pac and pPTOP3β plasmid. The *pac* gene (open box) is under the control of the 5'- and 3'-flanking regions of the glutamate dehydrogenase (*gdh*) gene (striated box). In construct pPTOP3β, the *top3β* gene is under the control of its own 5'-flanking region (open box) and the 3'-flanking region of the *ran* gene (dotted box). The filled black box indicates the coding sequence of the HA epitope tag. (*e*) TOP3β-HA level increased during encystation in TOP3β-overexpressing cells. The pPTOP3β stable transfectants were cultured in growth (Veg, vegetative growth) or encystation medium for 24 h (Enc, encystation) and then subjected to SDS-PAGE and Western blot analysis. The blot was probed with anti-HA and anti-RAN antibodies, respectively. Equal amounts of protein loading were confirmed by SDS-PAGE and Coomassie Blue staining. The RAN level slightly decreased during encystation. The ratio of TOP3β-HA protein over the loading control (Coomassie Blue-stained proteins) is calculated as described in figure 1*c*.

TOP3β with the HA tag significantly increased during encystation (figure 1*e*).

## 2.3. Change of localization of the TOP3β mutants

To further understand the function of *Giardia* TOP3β, we analysed the effect of mutation of TOP3β. The type I topoisomerases use an important Tyr of the cleavage domain as the active-site residue to create a transient single-stranded DNA break by transesterification [13]. We tried to understand whether Tyr 328 of TOP3β, which corresponds to Tyr 336 of the human TOP3β, is also important for its activity (figure 2*a*; electronic supplementary material, figure S1). Interestingly, the wild-type TOP3β-HA was located to the nuclear periphery

that partly overlapped with DAPI and slightly to the cytoplasm (figure 2*b*). The perinuclear staining pattern of TOP3β-HA is not the endoplasmic reticulum (ER) staining. The typical ER staining of *Giardia* as shown in BIP staining contains the reticulum shape of an interconnected network in the cytoplasm and slightly perinuclear staining that did not overlap with DAPI (electronic supplementary material, figure S3). Similarly, DNA topoisomerases I from yeast and human have been found to have a perinuclear distribution that may help function in DNA replication with perinuclear anchors of chromosomes [34]. Interestingly, the CWP1 protein was stained in the ESVs of TOP3β-HA positive stained cells (figure 2*c*), suggesting that TOP3β may function in inducing the ESV and thereby in inducing cyst formation. We found that mutation of the Tyr

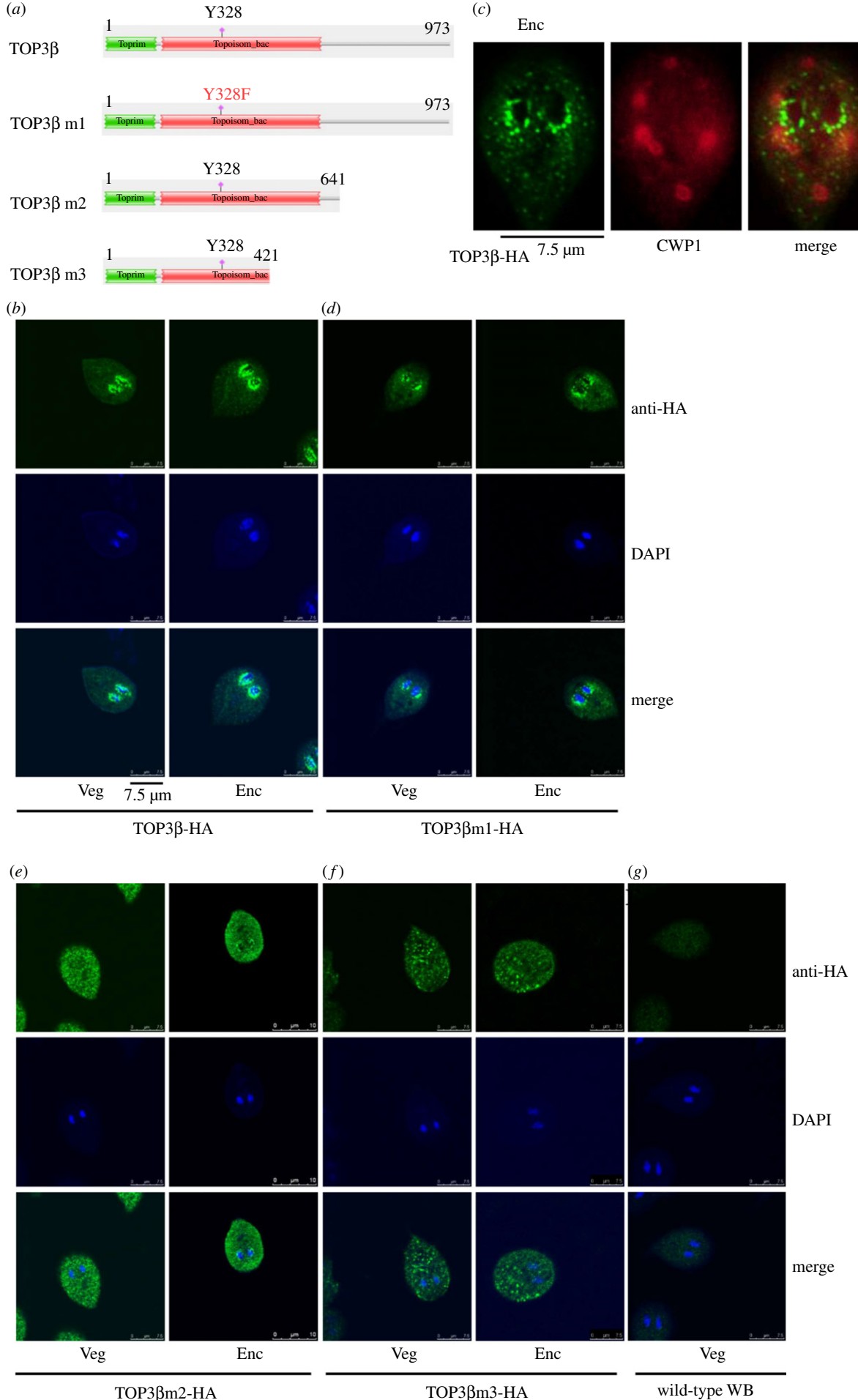

**Figure 2.** (*Caption opposite.*)

**Figure 2.** (*Opposite.*) Localization of TOP3β mutants. (*a*) Diagrams of TOP3β and TOP3βm1-3. The residue Tyr 328 (Y328), which is important for TOP3β activity, is mutated to Phe (F328) in TOP3βm1. TOP3βm2 remains the same as wild-type TOP3β, except that it does not contain the C-terminal zinc ribbon domain (deletion of residues 642–973). TOP3βm3 remains the same as wild-type TOP3β, except that it does not contain the C-terminal zinc ribbon domain and part of the Topoisom_bac domain (deletion of residues 422–973). The *top3* gene was mutated and subcloned to replace the wild-type *top3β* gene in the backbone of pPTOP3β (figure 1*d*), and the resulting plasmids pPTOP3βm1-3 were transfected into *Giardia*. The expression cassettes of the *pac* gene and *top3β* gene are the same as in figure 1*d*. (*b*) Perinuclear localization of the TOP3β protein. The pPTOP3β stable transfectants were cultured in growth (Veg, left panel) or encystation medium for 24 h (Enc, right panel), and then subjected to immuno-fluorescence analysis using anti-HA antibody for detection. The upper panels show that the TOP3β protein is localized to the nuclear periphery and slightly to the cytoplasm of vegetative and encysting trophozoites. The middle panels show the DAPI staining of cell nuclei. The bottom panels show the merged images. Some perinuclear staining of TOP3β-HA overlapped with DAPI. (*c*) Localization of CWP1 in the TOP3β-overexpressing cell line. The pPTOP3β stable transfectants were cultured in encystation medium for 24 h and then subjected to immunofluorescence assays. The endogenous CWP1 protein and vector-expressed TOP3β-HA protein were detected by anti-CWP1 and anti-HA antibodies, respectively. The left panel shows that the TOP3β-HA protein is localized to the nuclear periphery and slightly to the cytoplasm. The middle panel shows that the CWP1 protein is localized to the ESVs. The right panel shows the merged image. (*d–f*) Immunofluorescence analysis of TOP3βm1-3 distribution. The pPTOP3βm1-3 stable transfectants were cultured and then subjected to immunofluorescence analysis as described in figure 2*b*. The products of pPTOP3βm1 localized to the nuclear periphery that overlapped with DAPI with slight cytoplasmic staining in both vegetative and encysting trophozoites (*d*). The products of pPTOP3βm2 localized to the cytoplasm with minor presence in perinuclear region that overlapped with DAPI in both vegetative and encysting trophozoites (*e*). The products of pPTOP3βm3 localized to the vesicles in cytoplasm with minor presence in perinuclear region that overlapped with DAPI in both vegetative and encysting trophozoites (*f*). (*g*) Negative control for immunofluorescence. The wild-type WB trophozoites were cultured in growth (Veg, vegetative growth) and then subjected to immunofluorescence analysis using anti-HA antibody for detection as described in figure 2*b*.

328 to Phe did not change the localization of TOP3β to the nuclear periphery (TOP3βm1; figure 2*a,d*). We also found that deletion of the C-terminal 332 amino acids corresponding to the zinc ribbon domain (residues 642–973, pPTOP3βm2; figure 2*a,e*) resulted in a decrease in perinuclear localization, but an increase in cytosolic localization. Deletion of the C-terminal 552 amino acids corresponding to the zinc ribbon domain and a part of topoisomerase domain (residues 422–973, pPTOP3βm3; figure 2*a,f*) also decreased perinuclear localization, but increased localization to cytosolic vesicles. The background staining was very low as observed with wild-type WB trophozoites (figure 2*g*). TOP3βm2 and TOP3βm3 have lower but still some ability to localize to the nuclear periphery (figure 2*e,f*). The results suggest that the Zinc ribbon domain may play a partial role in the perinuclear localization.

## 2.4. Overexpression of TOP3β induced the expression of the *cwp1-3* and *myb2* genes

We further investigated the effect of the *Giardia* TOP3β on encystation. We found a significant increase in the CWP1 and MYB2 levels in the TOP3β-overexpressing cell line relative to the control cell line (figures 1*d* and 3*a*) [35]. The mRNA expression of the endogenous *top3β* plus vector-expressed *top3β* significantly increased in the TOP3β-overexpressing cell line relative to the control cell line (figure 3*b,c*). The mRNA expression of *cwp1-3* and *myb2* also increased in the TOP3β-overexpressing cell line (figure 3*b,c*). In previous studies, we obtained consistent cyst number data for *Giardia* growth stage due to spontaneous differentiation [36]. We found that the cyst number significantly increased in the TOP3β-overexpressing cell line (figure 3*d*). Similar results were obtained during encystation (electronic supplementary material, figure S3). These findings suggest that overexpression of TOP3β can increase expression of *cwp1-3* and *myb2* and cyst formation.

We further investigated the role of TOP3β by mutation analysis. We found that the levels of TOP3βm1 and TOP3βm2 were similar to that of wild-type TOP3β during vegetative growth, but TOP3βm3 was expressed at a lower level (figure 3*a*). We also found that the CWP1 level significantly decreased in the TOP3βm1- and TOP3βm3-expressing cell lines relative to the wild-type TOP3β-expressing cell line (figure 3*a*). The CWP1 level also significantly decreased in

the TOP3βm2-expressing cell line, but with a lower effect than in the TOP3βm1- and TOP3βm3-expressing cell lines (figure 3*a*). We further analysed whether the transcript levels were changed. As shown by RT-PCR analysis, the mRNA expression of *top3βm2-HA* and *top3βm3-HA* increased compared with that of wild-type *top3β-HA* during vegetative growth, but the mRNA expression of *top3βm1-HA* decreased (figure 3*b*). The mRNA expression of *cwp1-3* and *myb2* significantly decreased in the TOP3βm1-m3-expressing cell lines relative to the wild-type TOP3β-expressing cell line (figure 3*b,c*). The level of cyst formation significantly decreased in the TOP3βm1-m3-expressing cell lines relative to the wild-type TOP3β-expressing cell line (figure 3*d*). Similar results were obtained during encystation (electronic supplementary material, figure S3). The findings suggest a decrease in encystation-inducing activity of TOP3βm1-m3.

Oligonucleotide microarray assays confirmed upregulation of *cwp1-3* and *myb2* expression in the TOP3β-overexpressing cell line to approximately 1.5 to approximately 11.9-fold of the levels in the control cell line (figure 3*e*). The *ran* mRNA expression in the TOP3β-overexpressing cell line slightly decreased (approx. 0.8-fold; figure 3*e*). We found that 93 and 40 genes were significantly upregulated (greater than or equal to twofold) and downregulated (less than or equal to 1/2) ($p < 0.05$) in the TOP3β-overexpressing cell line relative to the vector control, respectively (electronic supplementary material, table S1). The *top3β* mRNA expression increased by approximately 2.1-fold ($p < 0.05$) in the TOP3β-overexpressing cell line (figure 3*e*).

## 2.5. TOP3β has DNA cleavage activity

The type I topoisomerases have ability to bind to and cleave single-stranded DNA [12,13,37]. *Drosophila* TOP3β cleaves DNA by forming a covalent topoisomerase–DNA complex [38]. To test DNA cleavage activity of TOP3β, we expressed TOP3β in *E. coli* and purified it to greater than 95% homogeneity. We performed DNA cleavage assays with purified recombinant TOP3β and pBluescript SK(+) plasmid. As shown in figure 4*a*, TOP3β has DNA cleavage activity.

Norfloxacin, a type II topoisomerase inhibitor, also inhibits *E. coli* topoisomerase I (type IA) at higher concentrations, resulting in anti-bacteria activity [25]. Norfloxacin can inhibit

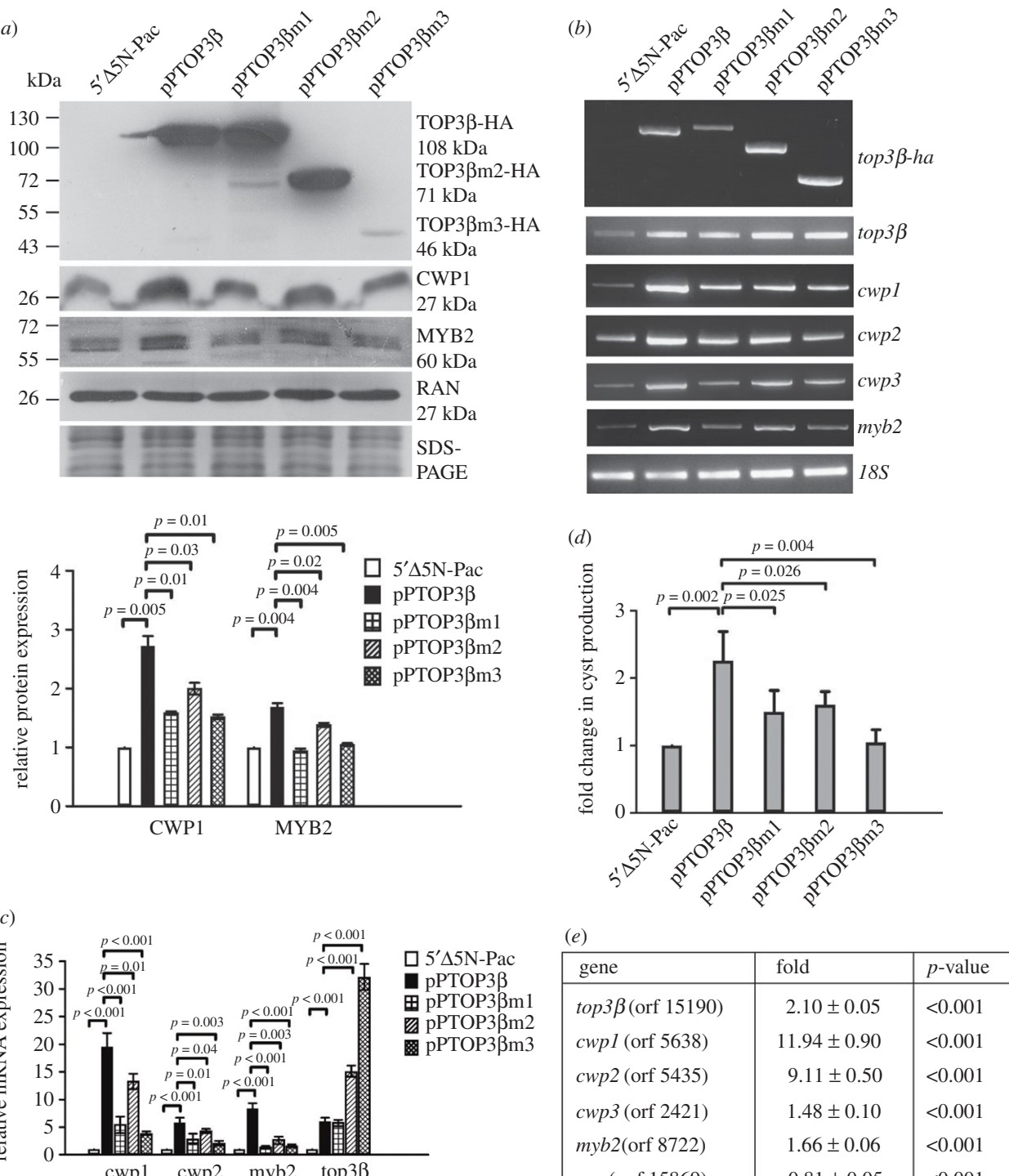

**Figure 3.** Induction of *cwp1-3* and *myb2* gene expression in the TOP3β-overexpressing cell line. (*a*) Overexpression of TOP3β increased the CWP1 and MYB2 levels. The 5′Δ5N-Pac, pPTOP3β, pPTOP3βm1, pPTOP3βm2 and pPTOP3βm3 stable transfectants were cultured in growth medium and then subjected to SDS-PAGE and Western blot. The blot was probed with anti-HA, anti-CWP1, anti-MYB2 and anti-RAN antibodies, respectively. Equal amounts of protein loading were confirmed by SDS-PAGE and Coomassie Blue staining. A similar level of the RAN protein was detected. The intensity of bands from three Western blot assays was quantified using ImageJ. The ratio of CWP1 and MYB2 proteins over the loading control RAN is calculated. Fold change is calculated as the ratio of the difference between the specific cell line and 5′Δ5N-Pac cell line, to which a value of 1 was assigned. Results are expressed as mean ± 95% confidence intervals. $p < 0.05$ was considered significant and the value was shown. (*b*) RT-PCR analysis of gene expression in the TOP3β- and TOP3β mutant-expressing cell lines. The 5′Δ5N-Pac, pPTOP3β and pPTOP3βm1-m3 stable transfectants were cultured in growth medium and then subjected to RT-PCR analysis using primers specific for *top3β-ha*, *top3β*, *cwp1*, *cwp2*, *cwp3*, *myb2* and 18S ribosomal RNA genes, respectively. Similar levels of the 18S ribosomal RNA for these samples were detected. (*c*) Quantitative real-time PCR analysis of gene expression in the TOP3β- and TOP3β mutant-expressing cell lines. Real-time PCR was performed using primers specific for *top3β*, *cwp1*, *cwp2*, *myb2* and 18S ribosomal RNA genes, respectively, as described in figure 1*b*. (*d*) TOP3β overexpression increased cyst formation. The pPTOP3β and pPTOP3βm1-m3 stable transfectants were cultured in growth medium and then subjected to cyst count as described under 'Material and methods'. The sum of total cysts is expressed as a relative expression level over control. Values are shown as means ± 95% confidence intervals. $p < 0.05$ was considered significant and the value was shown. (*e*) Microarray analysis. Microarray data were obtained from the 5′Δ5N-Pac and pPTOP3β cell lines during vegetative growth. Fold changes are shown as the ratio of transcript levels in the pPTOP3β cell line relative to the 5′Δ5N-Pac cell line. Results are expressed as the mean ± 95% confidence intervals of at least three experiments. $p < 0.05$ was considered significant and the value was shown.

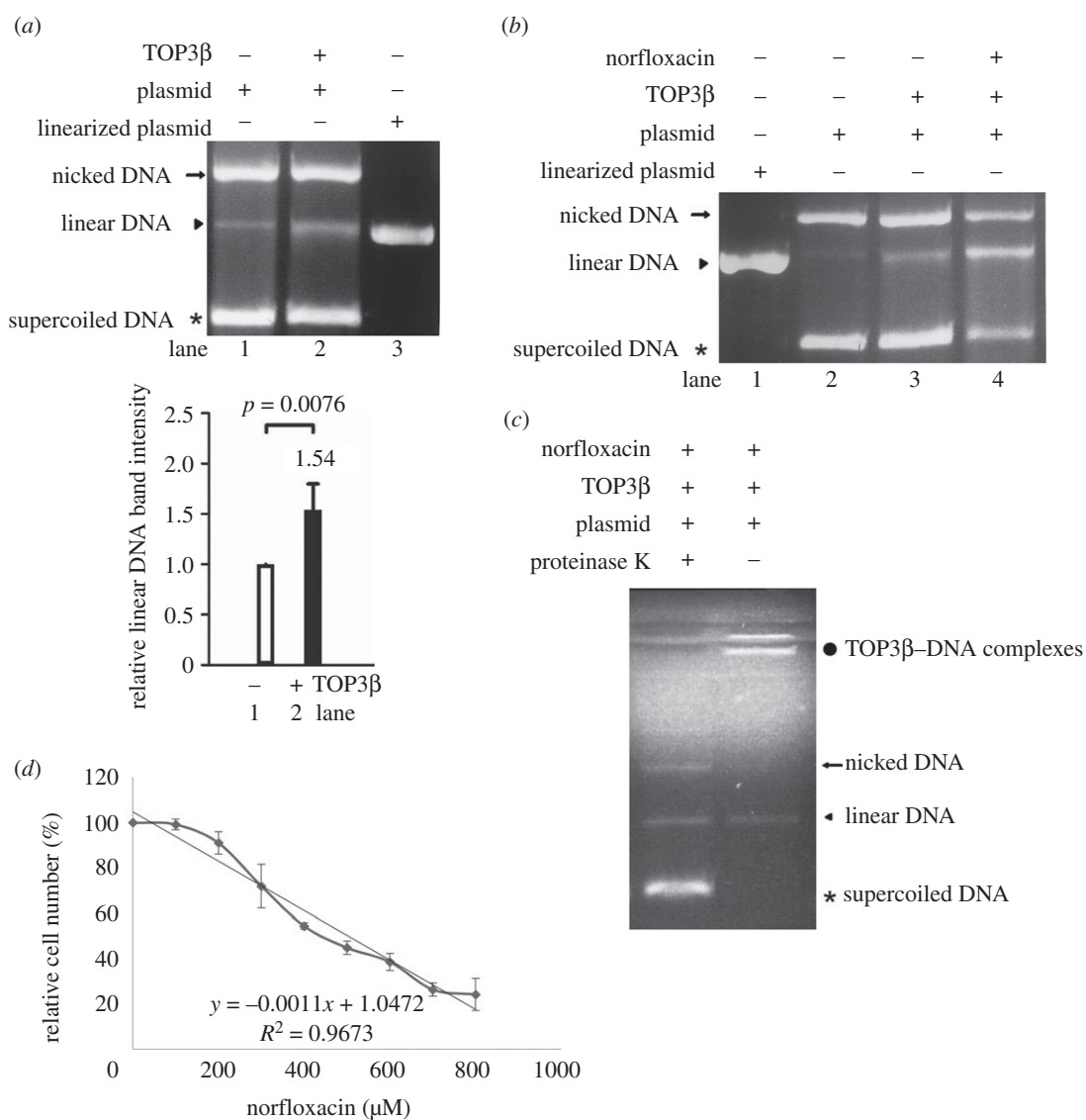

**Figure 4.** DNA cleavage activity of TOP3β and effect of norfloxacin. (*a*) TOP3β has DNA cleavage activity. DNA cleavage assays were performed with purified recombinant TOP3β and pBluescript SK(+) plasmid (3.0 kb). Components in the reaction are indicated above the lanes. Typically, 10 ng TOP3β was mixed with 300 ng plasmid DNA. Linearized plasmid was included as a size marker. The intensity of linear DNA bands from three assays was quantified using ImageJ. Fold change is calculated as the ratio of the '+ TOP3β' sample to the '− TOP3β' sample, to which a value of 1 was assigned. Results are expressed as mean ± 95% confidence intervals. $p < 0.05$ was considered significant and the value was shown. (*b*) Norfloxacin increased the cleavage complexes. DNA cleavage assays were performed with purified recombinant TOP3β and pBluescript SK(+) plasmid. Norfloxacin was added in the reaction as indicated above the lanes. Typically, 10 ng TOP3β was mixed with 300 ng plasmid DNA. Norfloxacin was dissolved in Me2SO. Adding Me2SO to the reaction mix was used as a control (lane 3). Adding 4.8 mM norfloxacin to the reaction mix increased the TOP3β DNA cleavage complexes (lane 4). Linearized plasmid was included as a size marker. (*c*) TOP3β formed covalent complexes with DNA. DNA cleavage assays were performed with purified recombinant TOP3β and pBluescript SK(+) plasmid. Norfloxacin was added in the reaction as indicated above the lanes. After reaction, proteinase K at a final concentration of 2 μg μl$^{-1}$ was added to the stop reaction of the cleavage assay, and then the products were analysed by agarose gel electrophoresis. The same volume of ddH2O was used for a negative reaction. (*d*) Anti-*Giardia* activity of norfloxacin. The wild-type non-transfected WB cells were subcultured at an initial density of $5 \times 10^4$ cells ml$^{-1}$ in growth medium containing 0, 100, 200, 300, 400, 500, 600, 700 or 800 μM norfloxacin for 24 h and then subjected to cell count. An equal volume of Me2SO was added to cultures as a negative control. The sum of total cells is expressed as a relative expression level over control. Values are shown as means ± 95% confidence intervals of three independent experiments.

topoisomerases by stabilizing covalent topoisomerase–DNA cleavage complexes [26]. To understand whether norfloxacin can inhibit *Giardia* TOP3β–DNA cleavage activity, we also performed DNA cleavage assays with norfloxacin. As shown in figure 4*b*, the addition of norfloxacin increased the amount of linear DNA, suggesting that norfloxacin can stabilize the TOP3β–DNA cleavage complex. We also tried to understand whether the products are from the covalent TOP3β–DNA cleavage complex. In a normal condition of the cleavage assay, proteinase K was included to stop the reaction by removing TOP3β from the cleavage complex (figure 4*c*). When proteinase K was not included, the TOP3β–DNA cleavage complex cannot enter the gel (figure 4*c*), suggesting that TOP3β can form a cleavage complex with DNA. The results indicate that TOP3β may function as a topoisomerase in *Giardia*.

## 2.6. Norfloxacin has anti-*Giardia* effect

Norfloxacin is an inhibitor of type IA and type II topoisomerases with anti-bacteria activity [25]. We found that norfloxacin increased DNA cleavage activity of TOP3β, indicating that norfloxacin can trap the cleavage complex

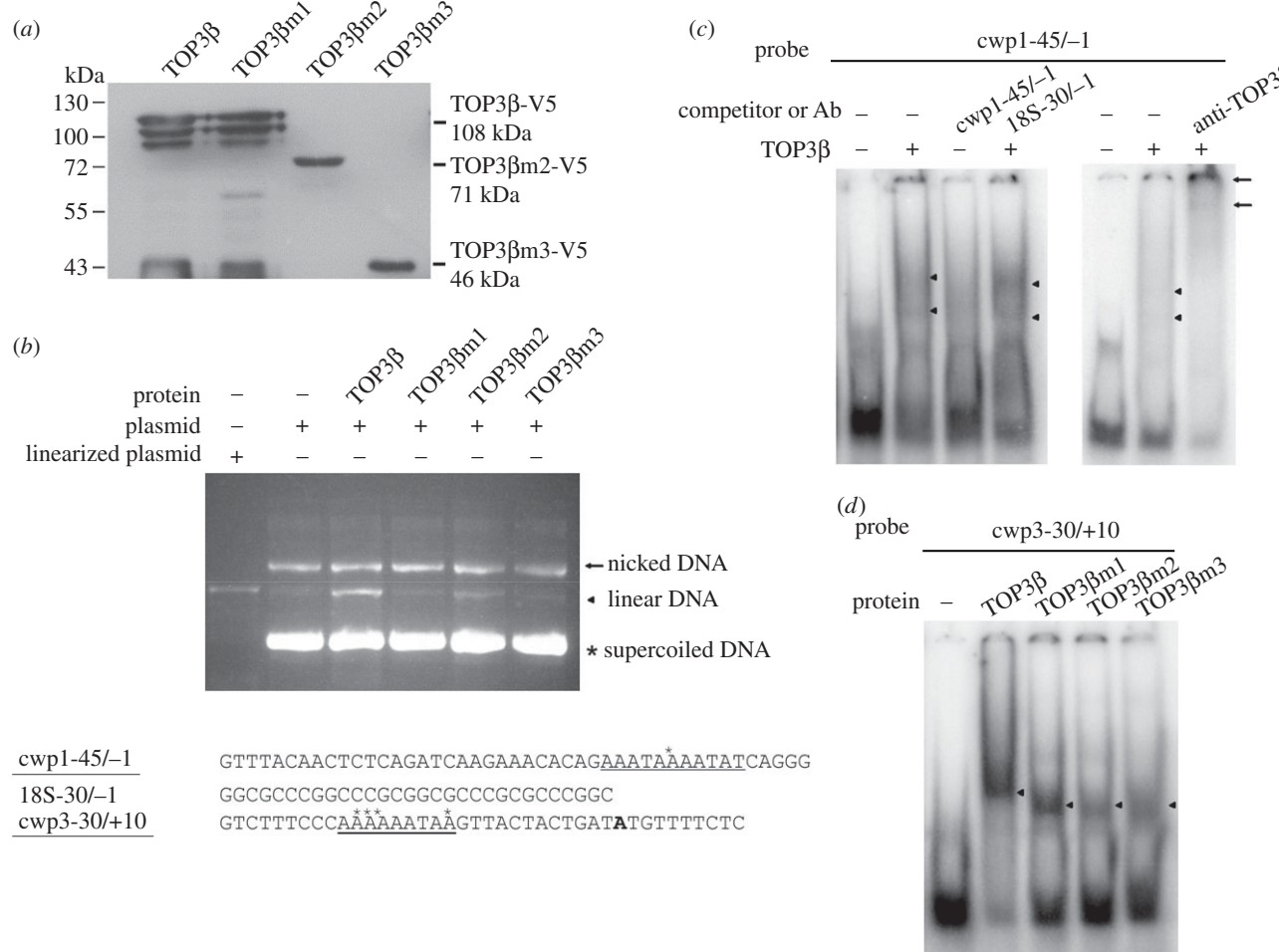

**Figure 5.** Decrease in DNA cleavage and DNA-binding activity of TOP3β mutants. (*a*) TOP3β and its mutants were purified from *E. coli* and detected by Western blot using anti-V5 antibody. (*b*) TOP3β mutants have lower DNA cleavage activity. DNA cleavage assays were performed with purified recombinant TOP3β and its mutants and pBluescript SK(+) plasmid. Typically, 10 ng TOP3β or its mutants and 300 ng plasmid DNA were used. Components in the reaction are indicated above the lanes. Linearized plasmid was included as a size marker. (*c*) DNA-binding ability of TOP3β. Electrophoretic mobility shift assays were performed using purified TOP3β and the $^{32}$P-end-labelled oligonucleotide probe cwp1-45/−1 (−45 bp to −1 bp relative to the translation start site of the *cwp1* gene). Components in the binding reaction mixtures are indicated above the lanes. The arrowheads indicate the shifted complexes. The TOP3β-binding specificity was confirmed by competition and supershift assays. Some reaction mixtures contained 200-fold molar excess of cold oligonucleotides or 0.8 μg of anti-TOP3β antibody as indicated above the lanes. The transcription start sites of the *cwp1* and *cwp3* genes are indicated by asterisks. The AT-rich initiator elements spanning the transcription start sites are underlined. (*d*) Decrease in DNA-binding activity of TOP3β mutants. Electrophoretic mobility shift assays were performed using purified TOP3β and its mutants and the $^{32}$P-end-labelled oligonucleotide probe cwp3-30/+10. The arrowheads indicate the shifted complexes.

of TOP3β (figure 4*b*). We also found that treatment with nor-floxacin significantly reduced *Giardia* trophozoites growth (figure 4*d*). The half-maximal inhibitory concentration (IC50) of norfloxacin on *Giardia* was 497 μM (figure 4*d*). The addition of 497 μM norfloxacin also decreased cyst formation by 67% (electronic supplementary material, figure S3). The results from the topoisomerase inhibitor nor-floxacin suggest that TOP3β may regulate *Giardia* growth and differentiation into cysts.

## 2.7. TOP3β mutants have a lower cleavage activity

To understand which regions are important for cleavage activity, the specific TOP3β mutants were expressed in *E. coli*, and purified (figure 5*a*), and tested by cleavage assays. We found a decrease in cleavage activity of TOP3βm1 (with a mutation of the catalytically important Tyr 328) and TOP3βm3 (with a deletion of C-terminal 552 amino acids) and slight decrease in cleavage activity of TOP3βm2 (with a deletion of the C-terminal 332 amino acids; figures 2*a* and 5*b*).

## 2.8. TOP3β has DNA-binding activity and its mutants have lower DNA-binding activity

We further tested DNA-binding activity of TOP3β. Electrophoretic mobility shift assays were performed with the purified TOP3β protein and double-stranded DNA sequences from the 5′-flanking region of the *cwp* genes. Incubation of a labelled double-stranded DNA probe, cwp1-45/−1, with TOP3β resulted in the formation of retarded bands (figure 5*c*). The binding specificity was confirmed by competition and super-shift assays (figure 5*c*). The formation of the shifted cwp1-45/−1 bands was competed by a 200-fold molar excess of unlabelled cwp1-45/−1, but not by the same excess of a non-specific competitor, 18S-30/−1 (figure 5*c*), suggesting that TOP3β did not bind to GC-rich sequence. The bound form on cwp1-45/−1 could be supershifted by an anti-TOP3β antibody (figure 5*c*). The results suggest that *Giardia* TOP3β can bind to the *cwp1* promoter (-45/−1 region). TOP3β was also shown to bind to the *cwp3* promoter, cwp3-30/+10 (figure 5*d*). To understand which regions of TOP3β are important for DNA binding,

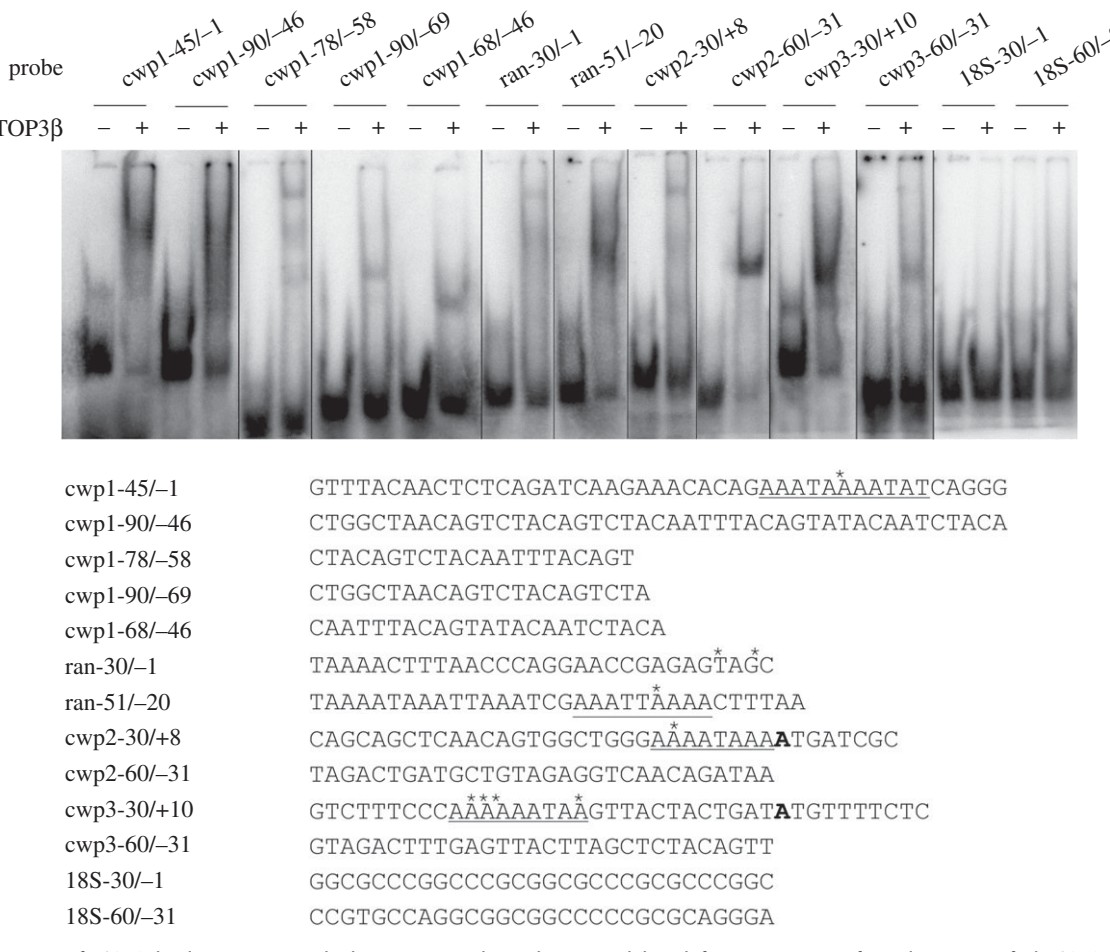

| probe | cwp1-45/−1 | cwp1-90/−46 | cwp1-78/−58 | cwp1-90/−69 | cwp1-68/−46 | ran-30/−1 | ran-51/−20 | cwp2-30/+8 | cwp2-60/−31 | cwp3-30/+10 | cwp3-60/−31 | 18S-30/−1 | 18S-60/−31 |

| cwp1-45/−1 | GTTTACAACTCTCAGATCAAGAAACACAG<u>AAATAĀAATAT</u>CAGGG |
| cwp1-90/−46 | CTGGCTAACAGTCTACAGTCTACAATTTACAGTATACAATCTACA |
| cwp1-78/−58 | CTACAGTCTACAATTTACAGT |
| cwp1-90/−69 | CTGGCTAACAGTCTACAGTCTA |
| cwp1-68/−46 | CAATTTACAGTATACAATCTACA |
| ran-30/−1 | TAAAACTTTAACCCAGGAACCGAGAGTĀGC |
| ran-51/−20 | TAAAATAAATTAAATCGAAATTĀAAACTTTAA |
| cwp2-30/+8 | CAGCAGCTCAACAGTGGCTGGGAĀAATAAA**A**TGATCGC |
| cwp2-60/−31 | TAGACTGATGCTGTAGAGGTCAACAGATAA |
| cwp3-30/+10 | GTCTTTCCCA<u>ĀĀĀAAATAĀ</u>GTTACTACTGAT**A**TGTTTTCTC |
| cwp3-60/−31 | GTAGACTTTGAGTTACTTAGCTCTACAGTT |
| 18S-30/−1 | GGCGCCCGGCCCGCGGCGCCCGCGCCCGGC |
| 18S-60/−31 | CCGTGCCAGGCGGCGGCCCCCGCGCAGGGA |

**Figure 6.** Detection of TOP3β binding sites in multiple promoters. Electrophoretic mobility shift assays were performed using purified TOP3β and $^{32}$P-labelled oligonucleotide probes. Components in the binding reaction mixtures are indicated above the lanes. The transcription start sites of the *cwp1*, *cwp2* and *cwp3* genes determined from 24 h encysting cells are indicated by asterisks. The AT-rich initiator elements spanning the transcription start sites are underlined. The translation start sites of the *cwp2* and *cwp3* genes are bold. '18S' represents 18S ribosomal RNA.

the specific mutants were tested for their DNA-binding activity. There was only a slight decrease in the DNA-binding activity of TOP3βm1, but there was a far more decrease in the DNA-binding activity of TOP3βm2 and TOP3βm3 (figure 5d).

TOP3β was also shown to bind to cwp1-90/−46, and within this region it weakly bound to the 5′-region (cwp1-90/−69), the middle region (cwp1-78/−58) or the 3′-region (cwp1-68/−46) (figure 6). We found that TOP3β bound strongly to the cwp2-60/−31 and cwp3-30/+10 probes, and weakly to the cwp2-30/+8 and cwp3-60/−31 probes (figure 6), suggesting that TOP3β can bind to other encystation-induced promoters, *cwp2* and *cwp3*. TOP3β also bound to a well-characterized *ran* core AT-rich promoter, ran-51/−20 [39] and weakly to ran-30/−1 (figure 6). TOP3β did not bind to the 18S-30/−1, and 18S-60/−31 probes, which do not contain AT-rich sequence (figure 6). Interestingly, TOP3β also weakly bound to a poly(A) sequence and a poly(A) sequence with a T or TT insertion (electronic supplementary material, figure S4), indicating that the TOP3β binding sequence contains AT-rich sequences. The results suggest that TOP3β can strongly bind to the *cwp1-3* and *ran* AT-rich promoter regions.

We also performed DNA-binding assays with single-stranded DNA probes. We found that TOP3β can bind to the single-stranded DNA of *cwp1-3* promoters (cwp1-45/−1F, cwp1-90/−46F, cwp2-60/−31F, cwp3-30/+10F) (electronic supplementary material, figure S5), but it cannot bind to the single-stranded DNA of 18S promoter (18S-30/−1F), which does not contain AT-rich sequence (electronic supplementary

material, figure S5). TOP3β bound to cwp3-30/+10F could be supershifted by an anti-TOP3β antibody (electronic supplementary material, figure S5). Both supershift results for the probes cwp1-45/−1 and cwp3 −30/+10F were significant as quantified in electronic supplementary material, figure S5. The results suggest that *Giardia* TOP3β can bind to the single-stranded DNA of the *cwp1-3* promoters.

## 2.9. Recruitment of TOP3β to the *top3β*, *cwp1-3* and *myb2* promoters

We further used norfloxacin-mediated topoisomerase immunoprecipitation assay [28,40], a method similar to ChIP assays, to study the association of TOP3β with the specific promoters. The addition of norfloxacin may increase the cleavage complex formation and thereby could increase ChIP sensitivity [40]. We found that TOP3β was associated with its own promoter and the *cwp1-3*, *myb2* and *ran* promoters during encystation (figure 7a,b). However, TOP3β was not associated with the U6 snRNA promoter (transcribed by pol III), nor the 18S ribosomal RNA promoter (transcribed by pol I), which has no TOP3β binding site (figure 7a).

## 2.10. Interaction between MYB2- and TOP3β-associated complexes

It is possible that TOP3β may regulate encystation-induced *cwp* genes by interacting with other transcription factors.

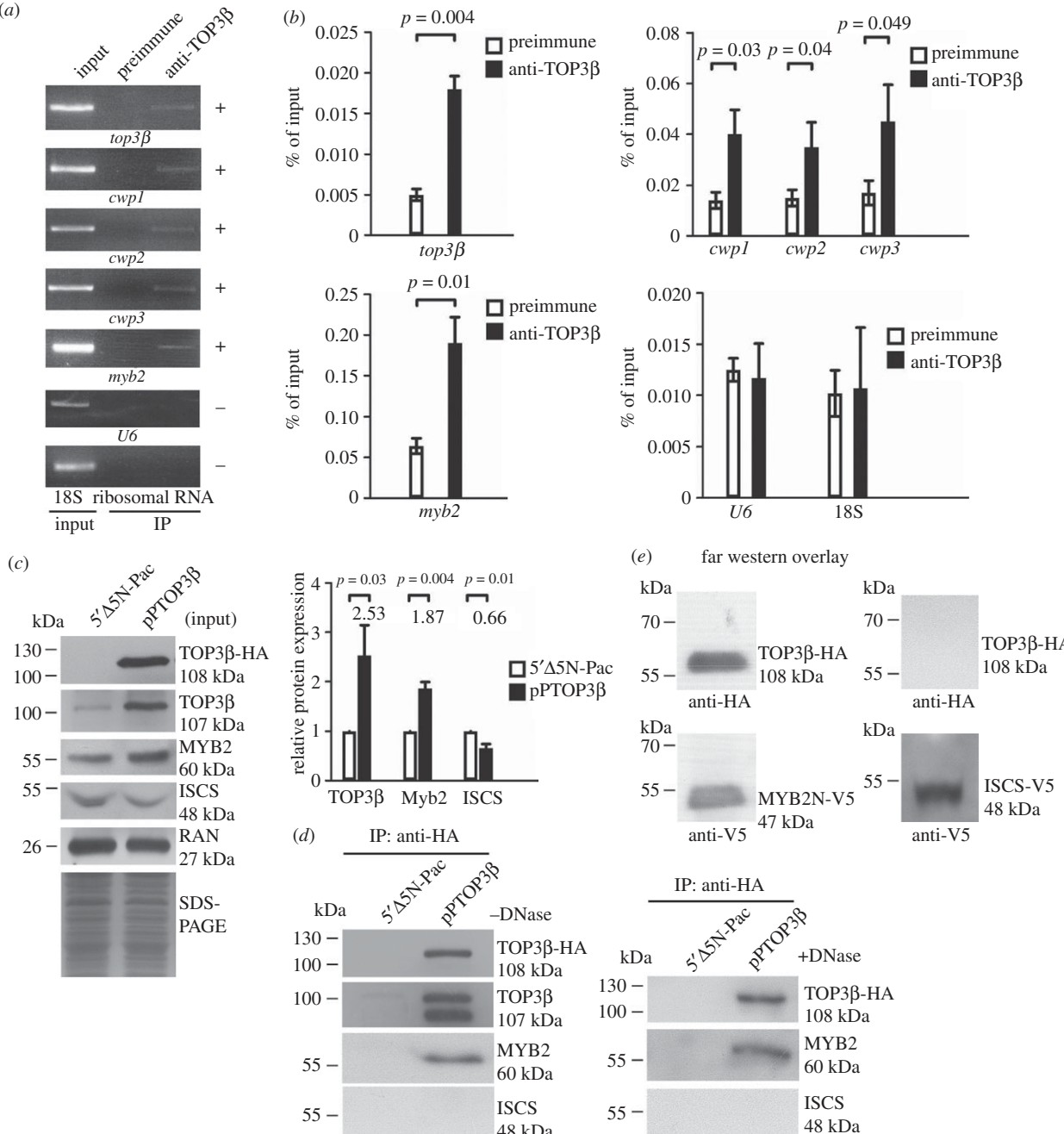

**Figure 7.** Recruitment of TOP3β to the *cwp* and *myb2* promoters and interaction between TOP3β and MYB2. (*a*) ChIP analysis of recruitment of TOP3β to the *cwp* and *myb2* promoters. The non-transfected WB cells were cultured in encystation medium containing 497 µM norfloxacin for 24 h and then subjected to norfloxacin-mediated topoisomerase immunoprecipitation assays. Anti-TOP3β was used to assess binding of TOP3β to endogenous gene promoters. Preimmune serum was used as a negative control. Immunoprecipitated chromatin was analysed by PCR using primers that amplify the 5′-flanking region of the specific genes. At least three independent experiments were performed. Representative results are shown. Immunoprecipitated products of TOP3β yield more PCR products of the *top3β*, *cwp1*, *cwp2*, *cwp3*, *myb2* gene promoters, indicating that TOP3β bound to these promoters (+). However, the anti-TOP3β antibody did not enrich the U6 promoter fragment (−). The 18S ribosomal RNA gene promoter was used as a negative control (−). (*b*) ChIP analysis coupled by quantitative PCR. Values represented as a percentage of the antibody-enriched chromatin relative to the total input chromatin (% of Input). Results are expressed as the mean ± 95% confidence intervals of at least three experiments. $p < 0.05$ was considered significant and the value was shown. (*c*) Expression of the TOP3β-HA, TOP3β, MYB2 and ISCS proteins detected in whole cell extracts for co-immunoprecipitation assays (Input). The 5′Δ5N-Pac and pPTOP3β stable transfectants were cultured in encystation medium for 24 h and then subjected to SDS-PAGE and Western blot analysis as described in figure 3a. The blot was probed with anti-TOP3β, anti-MYB2, anti-ISCS and anti-RAN antibodies, respectively. The intensity of bands from three Western blot assays was quantified as described in figure 3a. (*d*) Interaction between TOP3β and MYB2 detected by co-immunoprecipitation assays. The 5′Δ5N-Pac and pPTOP3β stable transfectants were cultured in encystation medium for 24 h. Proteins from cell lysates were immunoprecipitated using anti-HA antibody conjugated to beads. The precipitates were analysed by Western blotting with anti-HA, anti-TOP3β, anti-MYB2 and anti-ISCS antibodies, respectively, as indicated. (*e*) TOP3β and MYB2 interaction confirmed by Far Western blot analysis. Recombinant MYB2-N (residues 1–410) and ISCS proteins with a V5 tag at its C terminus was purified by affinity chromatography and detected by anti-V5 antibody in Western blot analysis (bottom panels). Far Western blot analysis was performed using the purified recombinant MYB2-N. The ISCS protein was used as a negative control. MYB2-N and ISCS were subjected to separation by SDS-PAGE, transferred onto a membrane, refolded in renaturation buffers, and incubated with lysate from the pPTOP3β stable transfectants as in figure 7d. Bound TOP3β-HA was detected with immunoblot using anti-HA antibody (upper panels). The purified recombinant MYB2-N and ISCS proteins on the membrane were detected using anti-V5 antibody (lower panels). TOP3β-HA bound to MYB2-N but not to ISCS.

royalsocietypublishing.org/journal/rsob    Open Biol. **10**: 190228

We further tried to understand whether TOP3β can interact with the encystation-induced MYB2 transcription factor [7]. We performed co-immunoprecipitation experiments using the TOP3β-overexpressing cell line. The TOP3β-HA protein (approx. 108 kDa) was expressed in the pPTOP3β stable cell line but not in the control cell line (5′Δ5N-Pac) (figure 7c) as detected by anti-HA antibody in Western blots (figure 7c). Overexpression of TOP3β in the pPTOP3β cell line also can be confirmed by the anti-TOP3β antibody (figure 7c). We found that TOP3β overexpression resulted in an increase in the MYB2 level (figure 7c). However, the ISCS level decreased by TOP3β overexpression (figure 7c). We lysed the cells and immunoprecipitated TOP3β-HA with anti-HA antibody. Western blots of immunoprecipitates probed with anti-HA and anti-MYB2 indicate that MYB2 co-precipitates with TOP3β-HA in the absence or the presence of DNase (figure 7d). The anti-HA antibody did not immunoprecipitate TOP3β-HA and MYB2 in the control cell line (figure 7d), nor did it immunoprecipitate ISCS in the pPTOP3β cell line (figure 7d). Far Western blot analysis confirmed the interaction between the N-terminal region of MYB2 (MYB2-N) and TOP3β-HA (figure 7e). The results suggest an interaction between MYB2 and TOP3β in a complex.

## 2.11. Targeted disruption of the *top3β* gene reduced expression of *cwp1-3* and *myb2*

To further understand the function of TOP3β, we analysed the effect of *top3β* gene disruption. We developed a CRISPR/Cas9 system to disrupt the *mlf* gene [11]. We further adapted this system to study the role of TOP3β. The CRISPR/Cas9 constructs were transfected into *G. lamblia* and TOP3βtd stable transfectants were established under puromycin selection (figure 8a). Scr7, an inhibitor of NHEJ, was added in the first replenishment of puromycin containing medium to increase knock-in efficiency via homologous recombination [11]. The replacement of the *top3β* gene with the puromycin acetyltransferase (*pac*) gene was confirmed by PCR and sequencing analysis of genomic DNA (figure 8b,c; electronic supplementary material, figure S6). The results show a successful disruption of the *top3β* gene by about 23% and a partial replacement of the *top3β* gene with the *pac* gene (figure 8b,c). It has been shown that G418 has cytotoxicity on mammalian cells [41]. The toxicity of G418 may be mediated by blocking polypeptide synthesis during translation elonglation [41]. Inhibition of protein synthesis further results in oxidative stress and cell death [41]. G418 also inhibits the growth of *Giardia* and can be used to select transfected cells [36]. We used G418 to test the drug sensitivity of the TOP3βtd cell line and found that TOP3βtd cell line exhibited increased sensitivity to it compared with the control cell line (figure 8d). We also found that the level of cyst formation significantly decreased in the TOP3βtd cell line relative to the control cell line during vegetative growth (figure 8e).

Western blot analysis confirmed the decrease of the TOP3β protein level in the TOP3βtd cell line relative to the control cell line (figure 8f). We found that the CWP1 level also significantly decreased in the TOP3βtd cell line relative to the control cell line (figure 8f). We further found the mRNA expression of *top3β*, *cwp1-3* or *myb2* significantly decreased in the TOP3βtd cell line relative to the control cell line (figure 8g; electronic supplementary material, figure S7). Similar results were obtained

during encystation (electronic supplementary material, figure S8). The findings suggest a decrease in the expression of *cwp1-3* and *myb2*, drug sensitivity and cyst formation by targeted disruption of the *top3β* gene.

We further tried to analyse results without puromycin. After selection, puromycin was removed from the TOP3βtd cell line to obtain the TOP3βtd –pu cell line. We found a successful disruption of the *top3β* gene by about 31% and a partial replacement of the *top3β* gene with the *pac* gene (electronic supplementary material, figure S9). The level of cyst formation significantly decreased in the TOP3βtd –pu cell line relative to the control –pu cell line (electronic supplementary material, figure S9). The levels of CWP1 and *top3β*, *cwp1-3* and *myb2* mRNA also significantly decreased in the TOP3βtd –pu cell line relative to the control –pu cell line (electronic supplementary material, figure S9). Similar results were obtained during encystation (electronic supplementary material, figure S10). The findings suggest a decrease in expression of *cwp1-3* and *myb2*, and cyst formation by targeted disruption of the *top3β* gene without puromycin.

## 3. Discussion

The type I topoisomerases are involved in cell growth, tissue development and cell differentiation [19–21]. In this study, we identified and characterized a type IA topoisomerase, TOP3β, from *Giardia*. TOP3β has DNA-binding and cleavage activity of topoisomerases (figures 4a and 5c), as its catalytically important domains and residues are conserved. This suggests that the type IA topoisomerases may have evolved before divergence of *Giardia* from the main eukaryotic line of descent. The presence of at least one type II topoisomerase and one type IA topoisomerase suggests that they play the necessary roles in different organisms [42]. Similarly, *Giardia* also has one type II topoisomerase and two type IA topoisomerases (orfs 16975, 15190 and 7615) [28].

Although mammalian topoisomerases have been studied intensively [18,43], information on how they function to transcriptional regulation is still emerging [44,45]. The type IA topoisomerases from *E. coli* and yeast can interact with RecQ DNA helicases to unwind hemicatenane structures during DNA replication [46], or unwind holliday junction during repairing DNA breaks or during chromosome segregation [43,47,48]. *Drosophila* type IB topoisomerases interacts with a splicing factor, SR protein, to regulate gene expression [13,49]. The C-terminal zinc ribbon domain of *E. coli* topoisomerase I (type IA) is important for interaction of RNA polymerase [18]. It helps bring to the transcription site for relaxation reaction [18], suggesting an importance of type I topoisomerases in transcription. The type IA topoisomerases may bind to DNA, regulate chromatin open and thereby activate gene expression [30,31,50]. Both type I and type II topoisomerases are recruited to genomic loci with higher transcriptional activity [45,51]. Human TOP3B targeted transcription start sites and induced transcription by repressing R-loop structures that inhibited transcription at target gene promoters [52,53]. We found that *Giardia* TOP3β may have a similar role in inducing transcription to help encystation. Our results show that *Giardia* TOP3β can bind to the specific sequences in the core AT-rich initiator promoter region of the genes encoding key components of the cyst wall, *cwp1-3* (figures 6 and 9; electronic supplementary material, figure

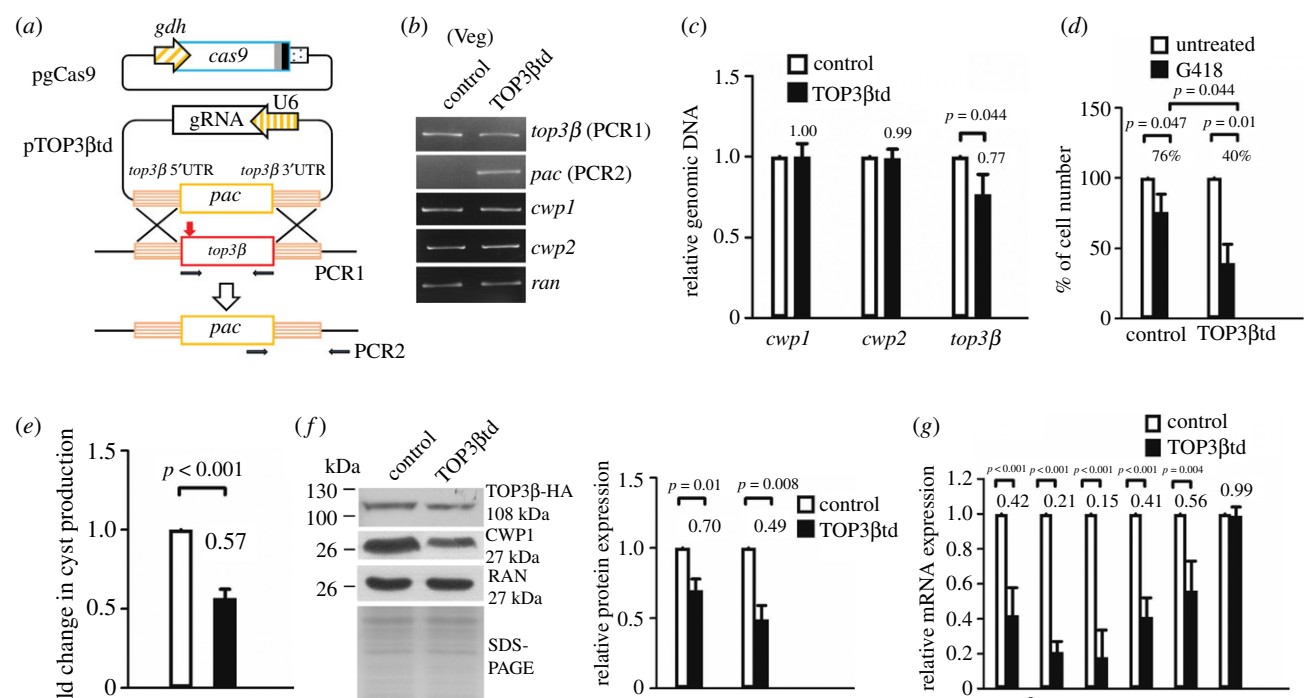

**Figure 8.** Decrease in expression of *cwp1-3* and *myb2* by targeted disruption of the *top3β* gene during vegetative growth. (*a*) Diagrams of the pgCas9 and pTOP3βtd plasmids. In construct pgCas9, the *cas9* gene is under the control of *gdh* promoter (striated box) and 3′ untranslated region of the *ran* gene (dotted box) and its product has a C-terminal nuclear localization signal (filled grey box) and an HA tag (filled black box). In construct pTOP3βtd, a single gRNA is driven by the *Giardia* U6 promoter. The single gRNA includes a guide sequence targeting 20-nucleotide of the *top3β* gene (nt 135–154), which is located upstream of three nucleotides of protospacer-adjacent motif (NGG sequence). pTOP3βtd also has the HR template cassette which contains the 5′ and 3′ flanking region of the *top3β* gene as homologous arms and the *pac* selectable marker. The Cas9/gRNA cutting site in the genomic *top3β* gene is indicated by a red arrow. After introducing a double-stranded DNA break in the *top3β* gene, replacement of the genomic *top3β* gene with the *pac* gene will occur by HR. The pgCas9 and pTOP3βtd constructs were transfected into *G. lamblia* WB trophozoites. An NHEJ inhibitor, SCR7, was added to increase HR. The TOP3βtd stable transfectants were established under puromycin selection. The control cell line is trophozoites transfected with double amounts of 5′Δ5N-Pac plasmid (figure 1*d*) and selected with puromycin. PCR1/2 were used for identification of clones with targeted disruption. (*b*) Partial replacement of the *top3β* gene with the *pac* gene in the TOP3βtd cell line confirmed by PCR. Puromycin was kept in the TOP3βtd and control cell lines. Genomic DNA was isolated from the TOP3βtd and control cell lines cultured in growth medium (vegetative growth, Veg). PCR was performed using primers specific for *top3β* (PCR1 in panel *a*), *pac* (PCR2 in panel *a*), *cwp1*, *cwp2* and *ran* genes, respectively. Products from the *cwp1*, *cwp2*, and *ran* genes are internal controls. (*c*) Partial disruption of the *top3β* gene in the TOP3βtd cell line confirmed by real-time PCR. Real-time PCR was performed using primers specific for *top3β*, *cwp1*, *cwp2* and *ran* genes, respectively. The *top3β*, *cwp1* and *cwp2* DNA levels were normalized to the *ran* DNA level. Fold changes in DNA levels are shown as the ratio of DNA levels in the TOP3βtd cell line relative to the control cell line. Results are expressed as the means ± 95% confidence intervals (error bars) of at least three separate experiments. $p < 0.05$ was considered significant and the value was shown. (*d*) Targeted disruption of the *top3β* gene increased G418 sensitivity. The TOP3βtd and control cell lines were subcultured at an initial density of $1 \times 10^6$ cells ml$^{-1}$ in growth medium containing 518 µM G418 for 24 h and then subjected to cell count. An equal volume of ddH2O was added to cultures as a negative control. The sum of total cells is expressed as a relative expression level over control. Values are shown as means ± 95% confidence intervals of three independent experiments. $p < 0.05$ was considered significant and the value shown. The viability of the TOP3βtd cell line decreased compared to the control cell line. (*e*) Cyst formation decreased by targeted disruption of the *top3β* gene in the TOP3βtd cell line during vegetative growth. The control and TOP3βtd cell lines were cultured in growth medium and then subjected to cyst count as described under 'Material and methods' and figure 3*d*. (*f*) Targeted disruption of the *top3β* gene decreased the CWP1 level in the TOP3βtd cell line during vegetative growth. The control and TOP3βtd cell lines were cultured in growth medium and then subjected to SDS-PAGE and Western blot analysis as described in figure 3*a*. The blot was probed with anti-TOP3β, anti-CWP1 and anti-RAN antibodies, respectively. The intensity of bands from three Western blot assays was quantified as described in figure 3*a*. (*g*) Decrease in expression of *cwp1-3* and *myb2* by targeted disruption of the *top3β* gene in the TOP3βtd cell line during vegetative growth. The control and TOP3βtd cell lines were cultured in growth medium and then subjected to quantitative real-time RT-PCR analysis using primers specific for *top3β*, *cwp1*, *cwp2*, *cwp3*, *myb2*, *ran* and 18S ribosomal RNA genes, respectively, as described in figure 1*b*.

S4). Similarly, *Drosophila* topoisomerase IIIβ prefers to bind to AT-rich DNA sequences [38]. We hypothesize that TOP3β, MYB2 and other transcription factors can bind to the AT-rich elements or the proximal upstream regions and form complexes (figure 9) [7]. This interaction may recruit RNA polymerase II to activate *cwp1-3* transcription (figure 9).

Many *Giardia* gene promoters have the AT-rich initiator elements responsible for promoter activity and transcription start site selection [39,54,55]. We have identified several transcription factors involved in the transactivation of the *cwp* genes, and they can bind to the AT-rich elements or the proximal upstream regions of the *cwp* promoters [7–10,56–58]. ChIP assays confirmed the binding of encystation-induced transcription factors E2F1 and MYB2 to the *cwp* and *myb2* gene promoters previously [7,10]. E2F1 and MYB2 may interact together to activate expression of the *cwp* genes [10]. We also found that MYB2 is co-immunoprecipitated with TOP3β (figure 7*d*). Treatment with DNase did not prevent the immunoprecipitation of Myb2 with TOP3β (figure 7*d*), suggesting that the interaction depends on

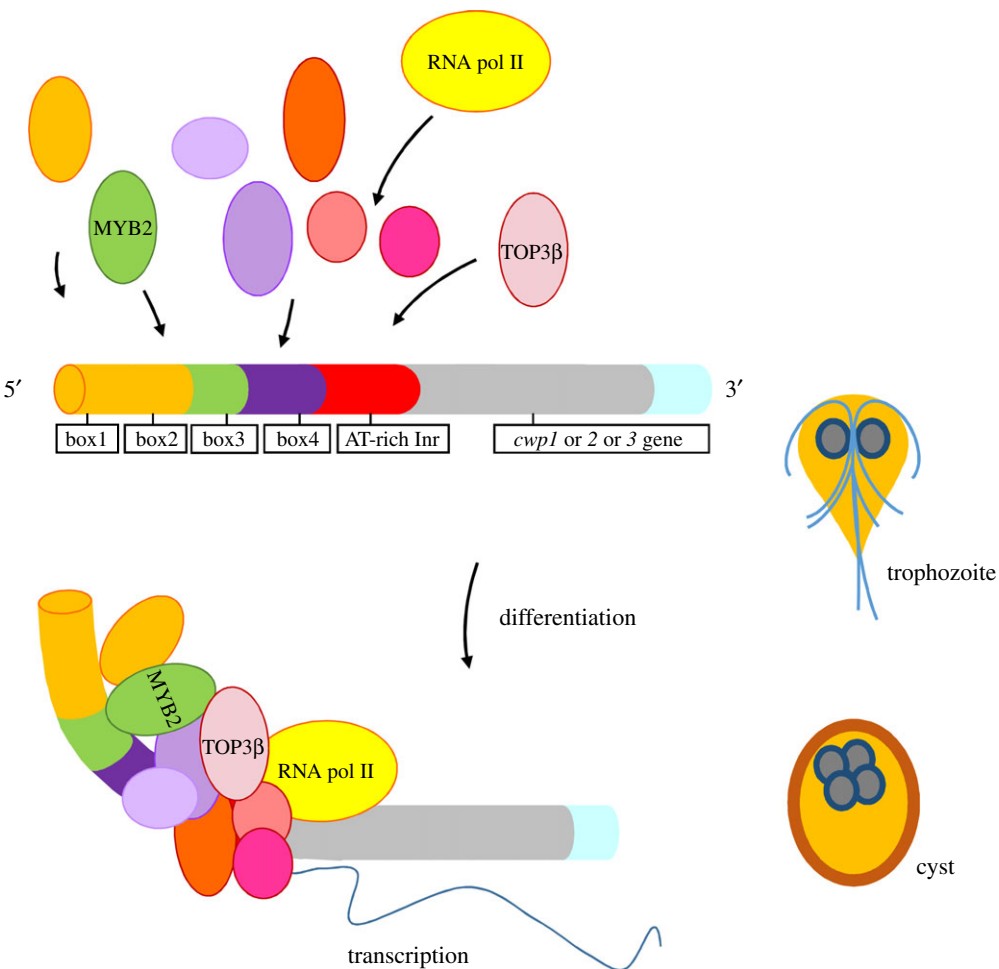

**Figure 9.** Increase of encystation-induced *cwp1–3* genes during differentiation into cysts. The genes encoding key components of the cyst wall, *cwp1–3*, are upregulated by TOP3β, MYB2 and other transcription factors during differentiation into cysts. These factors can bind to *cis*-acting elements, such as box1–4 or AT-rich initiator (Inr) of the *cwp1–3* promoter to activate *cwp1–3* transcription. TOP3β, MYB2 and other transcription factors, can form complexes and recruit RNA polymerase II to activate *cwp1–3* transcription. CWP1 was present in vegetative trophozoite stage at a lower level. During encystation, more CWP1 is produced by these factors. During encystation, the increase of MYB2, TOP3β and other transcription factors may further induce CWP1 expression, resulting in more cyst formation.

protein–protein interaction but not DNA. Far Western blot analysis, a non-antibody method, was further used to confirm this interaction between the MYB2-N and TOP3β-HA to avoid the non-specific problem (figure 7*e*). The MYB2-N can be phosphorylated by CDK2, which is involved in inducing encystation [6]. TOP3β can bind to AT-rich elements of both the constitutive *ran* gene and encystation-induced *cwp* genes (figure 6). However, overexpressed TOP3β can induce the CWP1 level but not the RAN level (figure 3*a*). This could be due to a lack of cooperation of the encystation-specific transcription factors to transactivate the constitutive *ran* gene. Similar results were found in studies of other transcription factors [7–10,57,58]. Interestingly, expression of all three TOP3β mutants (TOP3βm1-3) led to lower expression levels of CWP1, cyst formation, and *cwp1-3* and *myb2* mRNA relative to the wild-type TOP3β (figure 3*a–d*). However, it still leads to higher expression levels of CWP1, cyst formation, and *cwp1-3* and *myb2* mRNA relative to the vector control (figure 3*a–d*). It is possible that the mutants that were relatively overexpressed may still interact with transcription factors, such as Myb2, to activate expression of the *cwp* genes.

We also found that the *top3β* promoter contains the MYB2 binding sequences (electronic supplementary material, figure S10) [58], suggesting that *top3β* gene expression is upregulated by MYB2 and that TOP3β might play a positive role in *Giardia* encystation. Since overexpressed TOP3β increased the MYB2 level (figure 3*a*), there is a positive regulation cycle between TOP3β and MYB2. The human *top3α* promoters also contain the binding sequence of YY1 and USF1 activators, which are important for cell growth and differentiation [59]. The induction ability of the overexpressed TOP3β in *cwp* transcription was active in vegetative and encystation stages (figure 3; electronic supplementary material, figure S3). Similarly, Myb2 and Pax1 transcription factors in expression system can also induce *cwp* transcription in both stages [9,28]. This suggests that the specific promoter or enhancer elements for *cwp* transcription may be active in both stages.

Studies suggest that the C-terminal zinc ribbon domains of *E. coli* topoisomerase I and *Drosophila* topoisomerase IIIα are important for DNA binding [30,31]. The C-terminal zinc ribbon domain of *E. coli* topoisomerase I interacts with RNA polymerase to help bring to the transcription site for relaxation reaction [18]. We found that deletion of the C-terminal 332 amino acids (residues 642–973) corresponding to the zinc ribbon domain of TOP3β resulted in reduction in DNA-binding activity, but only slight decrease in cleavage activity (TOP3βm2) (figure 5*d*; electronic supplementary material, figure S1), suggesting that the zinc ribbon domain is important for DNA binding. Deletion of C-terminal 552 amino acids

(residues 422–973) corresponding to the zinc ribbon domain and a part of topoisomerase domain of TOP3β resulted in reduction in the cleavage activity (TOP3βm3; figures 1a and 5b), indicating that the topoisomerase domain is important for cleavage activity. We also found that a mutation of the catalytically important Tyr 328 resulted in a decrease in cleavage activity (TOP3βm1; figures 1a and 5b). Tyr327 of topoisomerase IIIβ in *Leishmania donovani* is also catalytically important [60]. Interestingly, mutation of the Tyr 328 to Phe did not change its perinuclear localization in both vegetative and encysting cells (TOP3βm1) (figure 2d). Mutation of this important Tyr also resulted in a significant decrease in the levels of CWP1, cyst formation, and *cwp1-3* and *myb2* mRNA (figures 3a–d and 5b,d), suggesting a correlation of DNA cleavage activity and *in vivo* function.

Typically, nuclear localization signal (NLS) is a region rich with basic amino acids. Two putative NLS motifs were predicted in TOP3β using the PSORT program (http://www.psort.org/), including RKHR at 970, and RRAAQPKRHGPRGRKHR at 957. We also found that deletion of the C-terminal 332 (residues 642–973) or 552 amino acids (residues 422–973) resulted in a decrease but not complete loss of perinuclear localization (TOP3βm2 or m3) (figure 2e,f), suggesting that the C-terminal zinc ribbon domain may play a partial role in the perinuclear localization and that other NLS motifs may be present in TOP3β. Deletion of the C-terminal 332 amino acids (TOP3βm2) resulted in a significant decrease in the levels of CWP1, cyst formation, and *cwp1-3* and *myb2* mRNA (figure 3a–d), but the effect is lower than the TOP3βm1 and TOP3βm3. As shown in figure 2e, TOP3βm2 still has some ability to localize to nuclear periphery. Interestingly, DNA cleavage activity was less affected in this mutant (TOP3βm2) (figure 5b), suggesting again a correlation of DNA cleavage activity and *in vivo* function. The results suggest that TOP3β may enhance the encystation-induced expression of *cwp1-3* and *myb2* through its cleavage activity.

We found that 93 and 40 genes were significantly upregulated and downregulated in the TOP3β-overexpressing cell line relative to the vector control, respectively (electronic supplementary material, table S1). We also found that chemosensitivity of the TOP3βtd cell line significantly increased by the addition of G418 (figure 8d), suggesting that TOP3β may affect many genes involved in cell growth to survive antibiotic stress.

Norfloxacin, which belongs to quinolones, is an inhibitor of the type II topoisomerases, including topoisomerase II (DNA gyrase) and topoisomerase IV [25]. It also inhibits type IA topoisomerases at higher concentrations [25]. Quinolones can stabilize the topoisomerase II–DNA complex and prevent religation of DNA, resulting in anti-topoisomerase activity [23,26,61]. A model suggests that quinolones can form complexes with DNA and topoisomerase IV, and create barriers to DNA replication [62]. We found that the addition of norfloxacin increased the cleavage complex formation of *Giardia* TOP3β (figure 4b), and significantly decreased cell growth and cyst formation (figure 4d; electronic supplementary material, figure S3). The IC50 of norfloxacin is 497 μM, which is similar to the IC50 for mammalian cells (470 μM for CT-26 cells) [63]. It is much higher than the IC50 of norfloxacin against *E. coli* (0.3 μM) and the IC50 for inhibiting *E. coli* topoisomerase I activity (135 μM) [25]. The difference of effective norfloxacin concentrations between *Giardia* and *E. coli* could be due to the variability of the overall sequences of *Giardia* TOP3β and

*E. coli* topoisomerase I, which is helpful for designing therapeutic drugs. Our results suggest that norfloxacin is less effective than the standard drug metronidazole with an IC50 of 2.1 μM [64,65].

In this study, we found that TOP3β can induce expression of CWP1 and MYB2 that are involved in encystation of the protozoan *Giardia*, suggesting that TOP3β may be functionally conserved and involved in regulation of gene expression and cell differentiation. Further work is necessary to find the norfloxacin derivatives that inhibit *Giardia* cyst formation and growth without harming human cells. Our study provides evidence for the important role of TOP3β in the differentiation of *Giardia* trophozoites into cysts, leading to a greater understanding of the evolution of the mechanism regulating cell differentiation and parasite transmission.

# 4. Material and methods

## 4.1. *Giardia lamblia* culture. Trophozoites *Giardia lamblia* culture

Trophozoites of *G. lamblia* WB, clone C6 (see ATCC 50803) (obtained from ATCC), were cultured in modified TYI-S33 medium [66]. Encystation was performed as previously described [67]. Briefly, trophozoites grown to late log phase in growth medium were harvested and encysted for 24 h in TYI-S-33 medium containing 12.5 mg ml$^{-1}$ bovine bile at pH 7.8 at a beginning density of $5 \times 10^5$ cells ml$^{-1}$. In experiments exposing *Giardia* vegetative trophozoites to norfloxacin, WB clone C6 trophozoites were cultured in growth medium at a beginning density of $5 \times 10^5$ cells ml$^{-1}$ with 0, 100, 200, 300, 400, 500, 600, 700, 800 or 497 μM norfloxacin. In experiments exposing *Giardia* vegetative trophozoites to G418, TOP3βtd and control cell line were cultured in growth medium at a beginning density of $1 \times 10^6$ cells ml$^{-1}$ with 518 μM G418.

## 4.2. Cyst count

Cyst count was performed on the stationary phase cultures (approx. $2 \times 10^6$ cells ml$^{-1}$) during vegetative growth as previously described [36]. The cells were subcultured in growth medium with suitable selection drugs at an initial density of $1 \times 10^6$ cells ml$^{-1}$. Cells seeded at this density became confluent within 24 h. Confluent cultures were maintained for an additional 8 h to ensure that the cultures were in stationary phase (at a density of approx. $2 \times 10^6$ cells ml$^{-1}$). Cyst count was performed on these stationary phase cultures. Cyst count was also performed on 24 h encysting cultures. Total cysts including both type I and II cysts [68] were counted in a hemacytometer chamber.

## 4.3. Isolation and analysis of the *top3β* gene

Synthetic oligonucleotides used are shown in electronic supplementary material, table S2. The *G. lamblia* genome database (http://www.giardiadb.org/giardiadb/) [5,69] was searched with the keyword 'topoisomerase' for annotated genes. This search detected one putative homologue for topoisomerase IIIβ (TOP3β) (XM_001709742.1, orf 15190 in the *G. lamblia* genome database). The TOP3β coding region with 300 bp of 5′-flanking region was cloned and the nucleotide sequence was determined. To isolate the cDNA

royalsocietypublishing.org/journal/rsob    Open Biol. **10**: 190228

of the *top3β* gene, we performed RT-PCR with the *top3β*-specific primers using total RNA from *G. lamblia*. For RT-PCR, 5 µg of DNase-treated total RNA from vegetative and 24 h encysting cells was mixed with oligo (dT)12-18 and random hexamers and Superscript II RNase H- reverse transcriptase (Invitrogen). Synthesized cDNA was used as a template in subsequent PCR with primers top3βF and top3βR. Genomic and RT-PCR products were cloned into pGEM-T easy vector (Promega) and sequenced (Applied Biosystems, ABI). Comparison of genomic and cDNA sequences showed that the *top3β* gene contained no introns.

## 4.4. Genomic DNA extraction, PCR and quantitative real-time PCR analysis

Synthetic oligonucleotides used are shown in electronic supplementary material, table S2. Genomic DNA was isolated from trophozoites using standard procedures [70]. For PCR, 250 ng of genomic DNA was used as a template in subsequent PCR. PCR analysis of *top3β* (XM_001709742.1, orf 15190), *cwp1* (U09330, orf 5638), *cwp2* (U28965, orf 5435), and *ran* (U02589, orf 15869) genes was performed using primers top3βF (PCR1F) and top3βR (PCR1R), PCR2F and PCR2R, cwp1F and cwp1R, cwp2F and cwp2R, ranF and ranR, respectively. For quantitative real-time PCR, SYBR Green PCR master mixture was used (Kapa Biosystems). PCR was performed using an Applied Biosystems PRISMTM 7900 Sequence Detection System (Applied Biosystems). Specific primers were designed for detection of the *top3β*, *cwp1*, *cwp2* and *ran* genes: top3βrealF and top3βrealR; cwp1realF and cwp1realR; cwp2realF and cwp2realR; ranrealF and ranrealR. Two independently generated stably transfected lines were made from each construct and each of these cell lines was assayed three separate times. The results are expressed as a relative expression level over control. Student's *t*-tests were used to determine statistical significance of differences between samples.

## 4.5. RNA extraction, RT-PCR and quantitative real-time PCR analysis

Synthetic oligonucleotides used are shown in electronic supplementary material, table S2. Total RNA was extracted from *G. lamblia* cell line during vegetative growth or encystation using TRIzol reagent (Invitrogen). For RT-PCR, 5 µg of DNase-treated total RNA was mixed with oligo (dT)12-18 and random hexamers and Superscript II RNase H⁻ reverse transcriptase (Invitrogen). Synthesized cDNA was used as a template in subsequent PCR. Semi-quantitative RT-PCR analysis of *top3β* (XM_001709742.1, orf 15190), *top3β-ha*, *cwp1* (U09330, orf 5638), *cwp2* (U28965, orf 5435), *cwp3* (AY061927, orf 2421), *myb2* (AY082882, orf 8722), *ran* (U02589, orf 15869) and 18S ribosomal RNA (M54878, orf r0019) gene expression was performed using primers top3β865F and top3β926R, top3β865F and HAR, cwp1F and cwp1R, cwp2F and cwp2R, cwp3F and cwp3R, myb2F and myb2R, ranF and ranR, 18SrealF and 18SrealR, respectively. For quantitative real-time PCR, SYBR Green PCR master mixture was used (Kapa Biosystems). PCR was performed using an Applied Biosystems PRISMTM 7900 Sequence Detection System (Applied Biosystems). Specific primers were designed for detection of the *top3β*, *top3β-ha*, *cwp1*, *cwp2*, *cwp3*, *myb2*, *ran* and 18S ribosomal RNA genes: top3βrealF and top3βrealR; top3βHAF and HAR;

cwp1realF and cwp1realR; cwp2realF and cwp2realR; cwp3realF and cwp3realR; myb2realF and myb2realR; ranrealF and ranrealR; 18SrealF and 18SrealR. Each primer pair was determined for amplification efficiency approximately 95% based on the slope of the standard curve. Two independently generated stably transfected lines were made from each construct and each of these cell lines was assayed three separate times. The results are expressed as a relative expression level over control. Student's *t*-tests were used to determine statistical significance of differences between samples.

## 4.6. Plasmid construction

Synthetic oligonucleotides used are shown in electronic supplementary material, table S2. All constructs were verified by DNA sequencing with a BigDye Terminator 3.1 DNA Sequencing kit and an Applied Biosystems 3100 DNA Analyzer (Applied Biosystems). Plasmid 5′Δ5N-Pac was a gift from Dr Steven Singer and Dr Theodore Nash [35]. Plasmid pgCas9 has been previously described [11]. To make construct pPTOP3β, the *top3β* gene and its 300 bp of 5′-flanking region were amplified with oligonucleotides top3βNF and top3βMR, digested with NheI and MluI, and cloned into NheI and MluI digested pPop2NHA [71]. To make construct pPTOP3βm1, the *top3β* gene was amplified using two primer pairs top3βm1F and top3βMR, and top3βm1R and top3βNF. The two PCR products were purified and used as templates for a second PCR. The second PCR also included primers top3βNF and top3βMR, and the product was digested with NheI and MluI and cloned into the NheI and MluI digested pPop2NHA [71]. To make construct pPTOP3βm2 or pPTOP3βm3, the *top3β* gene was amplified using primers top3βNF and top3βm2MR or top3βm3MR, digested with NheI and MluI, and cloned into NheI and MluI digested pPop2NHA [71].

The 620-bp 5′-flanking region of the *top3β* gene was amplified with oligonucleotides top3β5HF and top3β5NR, digested with *Hind*III/*Nco*I and cloned into *Hind*III/*Nco*I digested 5′Δ5N-Pac, resulting in TOP3β5. The 700-bp 3′-flanking region of the *top3β* gene was amplified with oligonucleotides top3β3XF and top3β3KR, digested with *Xba*I/*Kpn*I and cloned into *Xba*I/*Kpn*I digested TOP3β5, resulting in TOP3β53. We used gene synthesis services from IDT to obtain the fragment top3β-guide. The NCBI Nucleotide Blast search was used to avoid the potential off-target effects of guide sequence. The top3β-guide was digested with *Kpn*I/*Eco*RI and cloned into *Kpn*I/*Eco*RI digested TOP3β53, resulting in pTOP3βtd.

## 4.7. Transfection and Western blot analysis

Cells transfected with the pP series plasmids containing the *pac* gene were selected and maintained with 54 µg ml⁻¹ (100 µM) of puromycin as described [35,72]. For CRISPR/Cas9 system, *Giardia* trophozoites were transfected with plasmids pTOP3βtd and pgCas9, and then selected in 100 µM puromycin. The culture medium in the first replenishment contained 6 µM Scr7 and 100 µM puromycin. The TOP3βtd stable transfectants were established after selection. Stable transfectants were maintained at 100 µM puromycin and were further analysed by Western blotting, or DNA/RNA extraction. The replacement of the *top3β* gene with the *pac* gene was confirmed by PCR and sequencing. The control is *G. lamblia* trophozoites transfected with double amounts of 5′Δ5N-Pac plasmid and selected with puromycin. This kind of control was used

because of its same puromycin condition and plasmid amounts. Puromycin can induce *cwp* expression [36]. For the removal of puromycin experiments, puromycin was then removed from the medium for each stable cell line to obtain TOP3βtd –pu and control –pu cell lines. Subsequent analysis was performed after the removal of the drug for a month.

Western blots were probed with anti-V5-horseradish peroxidase (HRP) (Invitrogen), anti-HA monoclonal antibody (1/5000 in blocking buffer; Sigma), anti-TOP3β (1/10 000 in blocking buffer) (see below), anti-CWP1 (1/10 000 in blocking buffer) [7], anti-MYB2 (1/5000 in blocking buffer) [10], anti-α-tubulin (1/10 000 in blocking buffer) (T6199, Sigma), anti-ISCS (1/10 000 in blocking buffer), anti-BIP (1/10000 in blocking buffer), anti-RAN (1/10 000 in blocking buffer) [57,58] or preimmune serum (1/5000 in blocking buffer), and detected with HRP-conjugated goat anti-mouse IgG (1/5000; Pierce) or HRP-conjugated goat anti-rabbit IgG (1/5000; Pierce) and enhanced chemiluminescence (ECL) (Millipore).

## 4.8. Expression and purification of the recombinant TOP3β protein

The genomic *top3β*, *iscs* or *bip* gene was amplified using oligonucleotides top3βF and top3βR, iscsF and iscsR, or bipF and bipR, respectively. The product was cloned into the expression vector pET101/D-TOPO (Invitrogen) in frame with the C-terminal His and V5 tag to generate plasmid pTOP3β, pISCS or pBIP. To make the pTOP3βm1, pTOP3βm2 or pTOP3βm3 expression vector, the *top3β* gene was amplified using primers top3βF and top3βR and specific template, including pPTOP3βm1, pPTOP3βm2 (use top3βm2R as reverse primer) or pPTOP3βm3 (use top3βm3R as reverse primer), and cloned into the expression vector. The pBIP, pTOP3β, pTOP3βm1, pTOP3βm2 or pTOP3βm3 plasmid was freshly transformed into *E. coli* BL21 Star™ (DE3) (Invitrogen). An overnight pre-culture was used to start a 250-ml culture. *Escherichia coli* cells were grown to an A600 of 0.5, and then induced with 1 mM isopropyl-D-thiogalactopyranoside (IPTG) (Promega) for 4 h. Bacteria were harvested by centrifugation and sonicated in 10 ml of buffer A (50 mM sodium phosphate, pH 8.0, 300 mM NaCl) containing 10 mM imidazole and protease inhibitor mixture (Sigma). The samples were centrifuged and the supernatant was mixed with 1 ml of a 50% slurry of nickel-nitrilotriacetic acid Superflow (Qiagen). The resin was washed with buffer A containing 20 mM imidazole and eluted with buffer A containing 250 mM imidazole. Fractions containing BIP, TOP3β, TOP3βm1, TOP3βm2 or TOP3βm3 were pooled, dialysed in 25 mM HEPES pH 7.9, 20 mM KCl, and 15% glycerol and stored at −70°C. Protein purity and concentration were estimated by Coomassie Blue and silver staining compared with bovine serum albumin. BIP, TOP3β, TOP3βm1, TOP3βm2 or TOP3βm3 was purified to apparent homogeneity (greater than 95%).

For purification of ISCS, bacteria expressing pISCS were harvested by centrifugation and sonicated in 10 ml of buffer G (100 mM sodium phosphate, 10 mM Tris–Cl, 6 M Guanidine Hydrochloride, pH8.0) containing 10 mM imidazole and complete protease inhibitor cocktail (Roche). The samples were centrifuged and the supernatant was mixed with 1 ml of a 50% slurry of Ni-NTA superflow (Qiagen). The resin was washed with buffer B (100 mM sodium phosphate, 10 mM Tris–Cl, 8 M urea, pH8.0) and buffer C (100 mM sodium phosphate, 10 mM Tris–Cl, 8 M urea, pH6.3) and eluted with buffer E (100 mM sodium phosphate, 10 mM Tris–Cl, 8M urea, pH4.5). Fractions containing ISCS were pooled, dialysed in 25 mM HEPES pH 7.9, 40 mM KCl, and 15% glycerol, and stored at −70°C. Protein purity and concentration were estimated by Coomassie Blue and silver staining compared with bovine serum albumin. ISCS protein was purified to apparent homogeneity (greater than 95%).

## 4.9. Generation of anti-TOP3β, anti-ISCS or anti-BIP antibodies

Purified TOP3β, ISCS or BIP protein was used to generate rabbit polyclonal antibodies through a commercial vendor (Angene, Taipei, Taiwan).

## 4.10. Immunofluorescence assay

The pPTOP3β, pPTOP3βm1, pPTOP3βm2 or pPTOP3βm3 stable transfectants were cultured in growth medium under puromycin selection. Cells cultured in growth medium or encystation medium for 24 h were harvested, washed in phosphate-buffered saline (PBS), and attached to glass coverslips ($2 \times 10^6$ cells/coverslip) and then fixed and stained [73]. Cells were reacted with anti-HA monoclonal antibody (1/300 in blocking buffer; Sigma) and anti-mouse ALEXA 488 (1/500 in blocking buffer, Molecular Probes) was used as the detector. ProLong antifade kit with 4′,6-diamidino-2-phenylindole (Invitrogen) was used for mounting. TOP3β, TOP3βm1, TOP3βm2 or TOP3βm3 was visualized using a Leica TCS SP5 spectral confocal system.

## 4.11. Electrophoretic mobility shift assay

Double-stranded oligonucleotides specified throughout were 5′-end-labelled as described [39]. Binding reaction mixtures contained the components described [56]. Labelled probe (0.02 pmol) was incubated for 15 min at room temperature with 5 ng of purified TOP3β, TOP3βm1, TOP3βm2 or TOP3βm3 protein in a 20 μl volume supplemented with 0.5 μg of poly (dI-dC) (Sigma). Competition reactions contained 200-fold molar excess of cold oligonucleotides. In an antibody supershift assay, 0.8 μg of an anti-TOP3β antibody (see above) was added to the binding reaction mixture. The mixture was separated on a 6% acrylamide gel by electrophoresis.

## 4.12. DNA cleavage assays

Cleavage assays were performed as described previously [74]. Reaction was performed in a 25 μl mixture containing 10 mM Tris–HCl pH 7.5, 100 mM KCl, 5 mM MgCl$_2$, 30 μg ml$^{-1}$ BSA, 300 ng pBluescript SK(+) plasmid, and 10 ng purified TOP3β, TOP3βm1, TOP3βm2 or TOP3βm3. Some reactions contained 4.8 mM norfloxacin dissolved in Me2SO or Me2SO as a control to test the effect on cleavage activity of topoisomerases. After incubation at 37°C for 30 min, reaction was stopped by addition of 0.5% SDS, 10 mM EDTA, and 2 μg μl$^{-1}$ proteinase K and incubation at 37°C for 30 min. The resulting DNA was separated by electrophoresis on 1% agarose gels plus 25 μg ml$^{-1}$ ethidium bromide.

## 4.13. Microarray analysis

RNA was quantified by A260 nm by an ND-1000 spectro-photometer (Nanodrop Technology, USA) and qualitated by a Bioanalyzer 2100 (Agilent Technology) with an RNA 6000 Nano LabChip kit. RNA from the pPTOP3β cell line was labelled by Cy5 and RNA from the 5′Δ5N-Pac cell line was labelled by Cy3. 0.5 µg of total RNA was amplified by a Low RNA Input Quick-Amp labelling kit (Agilent Technologies) and labelled with Cy3 or Cy5 (CyDye, Agilent Technologies) during the *in vitro* transcription process. 0.825 µg of Cy-labelled cRNA was fragmented to an average size of about 50–100 nucleotides by incubation with fragmentation buffer at 60°C for 30 min. Correspondingly fragmented labelled cRNA was then pooled and hybridized to a *G. lamblia* oligonucleotide microarray (Agilent Technologies, USA) at 65°C for 17 h. After washing and drying by nitrogen gun blowing, microarrays were scanned with an Agilent microarray scanner (Agilent Technologies) at 535 nm for Cy3 and 625 nm for Cy5. Scanned images were analysed by Feature Extraction v. 10.5.1.1 software (Agilent Technologies), and image analysis and normalization software was used to quantify signal and background intensity for each feature; data were substantially normalized by the rank consistency filtering LOWESS method. All data are MIAME compliant and the raw data have been deposited in an MIAME (http://www.mged.org/Workgroups/MIAME/miame.html) compliant database (GEO) with accession number GSE109912.

## 4.14. Co-immunoprecipitation assay

The specific stable transfectants were cultured in encystation medium with puromycin ($5 \times 10^7$ cells in 45 ml medium) and harvested after 24 h in encystation medium with puromycin and washed in phosphate-buffered saline. Cells were lysed in luciferase lysis buffer (Promega) and protease inhibitor (Sigma) and then vortexed with glass beads. The cell lysates were collected by centrifugation and then incubated with anti-HA antibody conjugated to beads (Sigma). The beads were washed four times with luciferase lysis buffer (Promega). Finally, the beads were then resuspended in sample buffer and analysed by Western blotting and probed with anti-HA monoclonal antibody (1/5000 in blocking buffer; Sigma), TOP3β (1/10 000 in blocking buffer) (see above), anti-MYB2 or anti-ISCS (1/10 000 in blocking buffer) (see above), and detected with HRP-conjugated goat anti-mouse IgG (Pierce, 1/5000) or HRP-conjugated goat anti-rabbit IgG (Pierce, 1/5000) and ECL (GE Healthcare).

## 4.15. Far Western blot analysis

The Myb2-N and ISCS with a V5-tag at C terminus were expressed in *E. coli* and purified as previously described [6] and as above, respectively. The MYB2-N-V5 and ISCS-V5 were resolved by SDS-PAGE, transferred onto PVDF membranes, refolded in renaturation buffers, and incubated with lysate from the pPTOP3β cell line as described in co-immuno-precipitation assays. Bound TOP3β-HA was detected with monoclonal anti-HA antibody (Sigma). The signal was detected with HRP-conjugated goat anti-mouse IgG (GE Healthcare) and ECL (GE Healthcare). Additional membranes for the resolved MYB2-N-V5 and ISCS-V5 proteins were incubated with anti-V5-HRP antibody and detected by ECL (GE Healthcare) to determine where the V5-tagged proteins migrated.

## 4.16. Norfloxacin-mediated topoisomerase immunoprecipitation assays

The WB clone C6 cells were inoculated into encystation medium containing 497 µM norfloxacin ($5 \times 10^7$ cells in 45 ml medium) and harvested after 24 h and washed in phosphate-buffered saline. The assay was performed as previously described [28] with some modifications. Formaldehyde was then added to the cells in phosphate-buffered saline at a final concentration of 1%. Cells were incubated at room temperature for 15 min and reactions were stopped by incubation in 125 mM glycine for 5 min. After phosphate-buffered saline washes, cells were lysed in luciferase lysis buffer (Promega) and protease inhibitor (Sigma) and then vortexed with glass beads. The cell lysate was sonicated on ice with Biorupter Plus (Diagenode) and then centrifuged. Chromatin extract was incubated with protein G plus/protein A-agarose (Merck) for 1 h. After the removal of protein G plus/protein A-agarose, the precleared lysates were incubated with 2 µg of anti-TOP3β antibody or preimmune serum for 2 h and then incubated with protein G plus/protein A-agarose (Merck) for 1 h. The beads were washed with luciferase lysis buffer (Promega) twice and phosphate-buffered saline twice. The beads were resuspended in elution buffer containing 50 mM Tris–HCl, pH 8.0, 1% SDS and 10 mM EDTA at 65°C for 4 h. To prepare DNA representing input DNA, 2.5% of precleared chromatin extract without incubation with anti-TOP3β was incubated with elution buffer at 65°C for 4 h. Eluted DNA was purified by the QIAquick PCR purification kit (Qiagen). Purified DNA was subjected to PCR reaction followed by agarose gel electrophoresis. Primers 18S5F and 18S5R were used to amplify the 18S ribosomal RNA gene promoter as a control for our analysis. Primers top3β5F and top3β5R, cwp15F and cwp15R, cwp25F and cwp25R, cwp35F and cwp35R, myb25F and myb25R, and U65F and U65R, were used to amplify *top3β*, *cwp1*, *cwp2*, *cwp3*, *myb2* and U6 gene promoters within the −200 to −1 region.

Data accessibility. This article has no additional data.

Authors' contributions. S.-C.W. and J.-H.W. conceived, designed and performed the experiments. S.-Y.T., L.-H.S. and M.-H.L. performed the experiments. G.A.L. wrote a paper outline. C.-H.S. performed the experiments, analysed the data and wrote the manuscript.

Competing interests. We declare we have no competing interests.

Funding. This work was supported by the National Science Council grant nos. MOST 99-2320-B-002-017-MY3, 100-2325-B-002-039, 101-2325-B-002-036-, 103-2628-B-002-006-MY3-, and 106-2320-B-002-038-MY2, and the National Health Research Institutes grant no. NHRI-EX99-9510NC in Taiwan, and was also supported in part by the Aim for the Top University Program of National Taiwan University, grant no. 33474.

Acknowledgements. We thank Ms. Yi-Li Liu and I-Ching Huang for technical support in DNA sequencing, Dr Tsai-Kun Li and Dr Nei-Li Chan for helpful comments, and Bo-Shiun Yan for kind support in providing laboratory facilities. We thank the staff of the cell imaging core at the First Core Labs, National Taiwan University College of Medicine, for technical assistance. We are also very grateful to the researchers and administrators of the *G. lamblia* genome database for providing genome information.

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
