## [Reviewer comments · Open Biology]

Review History

RSOB-19-0228.R0 (Original submission)

Review form: Reviewer 1

Recommendation

Accept with minor revision (please list in comments)

Do you have any ethical concerns with this paper?

No

Comments to the Author

Sun et al explore the role of TOP3 β in regulating *Giardia* differentiation. They demonstrate that TOP3 β localizes to the nucleus and binds to AT rich promoter sequences. They also show that over expression of this enzyme leads to up-regulation of CWP proteins and Myb2. Conversely knockdown with CRISPR based allele deletion results in reduced CWP and Myb2 expression. Overall the results are clearly presented. I reviewed an earlier version of this paper and the authors addressed my major criticisms, so I recommend acceptance with some minor revisions I will outline below.

Major Points

1. The change in localization observed for TOP3 β in non-encysting cells (Fig 1) versus encysting cells in Sup Fig 3 is fascinating and this deserves more attention. That TOP3 β localization changes from perinuclear to dispersed in the nucleus and is an important point that supports the authors assertion that TOP3 β is involved in regulating differentiation. This same re-localization is not observed in Fig 1 perhaps because the cell is not at the right stage of encystation or is not actually encysting (no CWP1 staining here). The cell shown in Fig S3 is sort of mid-encystation based on the CWP1 staining. So I also wonder if the nuclear distribution changes with stage in the encystation response. This is potentially a really nice observation that if moved to Fig 1 will strengthen the paper because it shows TOP3 β is specifically changing localization in actively differentiating cells. I would like the authors to include this result in Fig 1 and also quantify the number of cells that have dispersed nuclear versus perinuclear localization during encystation. It is also interesting that the mutants are more clearly nuclear localized during encystation. Since there is no quantification of this I don't know if this is due to selected images or a real response to encystation stimulus.

2. The super-shift assay in Fig 5C is not very convincing. A similar experiment in Fig S5B on the other hand is very clear. I believe the authors, but this experiment should be repeated and hopefully a clearer result can be obtained. What also doesn't make sense is that in S5B there are two bands one that is found in every lane and then a unique shifted band near the top of the blot in the presence of the antibody. In Fig 5C the same band is observed in every well although it does appear darker in the lane where the antibody was labeled. As an alternative to repeating the blot, the authors could quantify the intensity of this band versus the same position in the control lanes to convince us that there is enrichment in their replicates.

3. Regarding the CRISPR experiment I question the value of using CRISPR/CAS9 over homologous recombination. I am under the impression that CRISPR/CAS9 is not actually providing any benefit and that the authors are observing homologous recombination. After transfection we expect the provided repair template to enter one of the two nuclei based on Poxleitner et al 2008 and Carpenter et al 2012. Therefore, a single transfection should be able to edit two copies of the genome. It is not expected to generate a true knockout due to the two nuclei challenge. Here however, PCR (8B) and western blotting (8E) both point toward just a single allele being disrupted since genomic DNA and protein levels are pretty close to the 25% reduction level of losing a single allele. The authors mention the use of NHEJ inhibiting drugs which seem to be doing nothing since Ebnetter et al showed that homologous recombination can be used to knockout genes in *Giardia*. I wonder why the authors bother to invoke the use of CRISPR since there is no perceivable benefit? Note that I do not dispute their mutant has a phenotype. My concern is that their pseudo use of CRISPR/CAS9 will confuse readers.

Minor Points

1. Fig S1 is corrupted or there was an error in its construction. The region around T328 is not shown and neither is the red arrow that is mentioned by the legend. Please fix this.

2. Sentence on Line 157 is confusing. It reads as if it is interesting that ESVs contain CWP1. To me what is interesting is that the localization of TOP3 β is more dispersed in this encysting cell.

3. Sentence on 342 is awkward please re-write

4. On 411 change catalytic to catalytically

5. I would change the last sentence of the conclusion. I don't believe this study told us much about the evolution of TOP3 β and whether TOP3 β is really a viable drug target. The function of the gene wasn't shown to be essential since only a single allele of TOP3 β was deleted according to the PCR in Fig 8B which roughly corresponds to the 30% reduction in TOPO3 β protein levels. My recommendation: "Our study provides evidence for the important role of TOP3 β in the

differentiation of *Giardia* trophozoites into cysts, leading to greater understanding of the mechanism regulating cell differentiation and parasite transmission.”

Review form: Reviewer 2

Recommendation

Major revision is needed (please make suggestions in comments)

Do you have any ethical concerns with this paper?

No

Comments to the Author

The authors have submitted a detailed and interesting study of the effect of DNA topoisomerase 3b (type 1) on encystation in *Giardia*. The title focuses on the effect of cyst generation related to *cwp* expression. However, I'm not convinced that there is enough causal information to warrant the use of “by inducing” in the title.

They have used transfection studies to study the effect of upregulation of TOP3b on expression of the *cwps* and *myb2* on encystation. They have also used CRISPR/Cas9 to reduce expression of *top3b* and shown decreases in the above. (Of note, knockout is impossible with *Giardia* because of its polyploidy, so these approaches are reasonable).

Overall, the scientific component of the work seems solid, but I do have some questions regarding the data to be addressed by the authors:

Thus, I will begin with the results section:

For figure 1 and other figures, the authors simply use $p < 0.05$, but unless < 0.001 , it would be helpful to see the numbers for CI and actual p value.

For Fig 4A, the difference between *top3b* positive and negative for linear DNA is not very convincing. Can the authors provide quantitative scan data from several runs?

Fig 5C, the mobility shifted lanes are not very convincing.

The bigger issues for correction relate to the writing:

1. The introduction has far too much of results in it. Almost 1/3 of the intro is results.
2. The biggest problem is the discussion. I felt like I was reading the results again when reading the discussion. There are two things that would greatly help. First, rather than reiterating results, explain the difficult parts and show how the results fit into other literature. Second, a figure with a model of what the authors think is happening would be very useful.
3. Although the writing is reasonably good, there are still occasional grammatical errors. If the journal has in-house copy editing, it should be ok.
4. A structured abstract would be nice if it fits with journal policy.

Decision letter (RSOB-19-0228.R0)

12-Nov-2019

Dear Dr Sun,

We are writing to inform you that the Editor has reached a decision on your manuscript RSOB-

19-0228 entitled "DNA topoisomerase III β promotes cyst generation by inducing cyst wall protein gene expression in *Giardia lamblia*", submitted to Open Biology.

As you will see from the reviewers' comments below, there are a number of criticisms that prevent us from accepting your manuscript at this stage. The reviewers suggest, however, that a revised version could be acceptable, if you are able to address their concerns. If you think that you can deal satisfactorily with the reviewer's suggestions, we would be pleased to consider a revised manuscript.

The revision will be re-reviewed, where possible, by the original referees. As such, please submit the revised version of your manuscript within four weeks. If you do not think you will be able to meet this date please let us know immediately.

When submitting your revised manuscript, please respond to the comments made by the referee(s) and upload a file "Response to Referees" in "Section 6 - File Upload". You can use this to document any changes you make to the original manuscript. In order to expedite the processing of the revised manuscript, please be as specific as possible in your response to the referee(s).

Please see our detailed instructions for revision requirements
<https://royalsociety.org/journals/authors/author-guidelines/>

Sincerely,
The Open Biology Team
<mailto:openbiology@royalsociety.org>

Reviewer(s)' Comments to Author(s):

Referee: 1
Comments to the Author(s)

Sun et al explore the role of TOP3 β in regulating *Giardia* differentiation. They demonstrate that TOP3 β localizes to the nucleus and binds to AT rich promoter sequences. They also show that over expression of this enzyme leads to up-regulation of CWP proteins and Myb2. Conversely knockdown with CRISPR based allele deletion results in reduced CWP and Myb2 expression. Overall the results are clearly presented. I reviewed an earlier version of this paper and the authors addressed my major criticisms, so I recommend acceptance with some minor revisions I will outline below.

Major Points

1. The change in localization observed for TOP3 β in non-encysting cells (Fig 1) versus encysting cells in Sup Fig 3 is fascinating and this deserves more attention. That TOP3 β localization changes from perinuclear to dispersed in the nucleus and is an important point that supports the

authors assertion that TOP3 β is involved in regulating differentiation. This same re-localization is not observed in Fig 1 perhaps because the cell is not at the right stage of encystation or is not actually encysting (no CWP1 staining here). The cell shown in Fig S3 is sort of mid-encystation based on the CWP1 staining. So I also wonder if the nuclear distribution changes with stage in the encystation response. This is potentially a really nice observation that if moved to Fig 1 will strengthen the paper because it shows TOP3 β is specifically changing localization in actively differentiating cells. I would like the authors to include this result in Fig 1 and also quantify the number of cells that have dispersed nuclear versus perinuclear localization during encystation. It is also interesting that the mutants are more clearly nuclear localized during encystation. Since there is no quantification of this I don't know if this is due to selected images or a real response to encystation stimulus.

2. The super-shift assay in Fig 5C is not very convincing. A similar experiment in Fig S5B on the other hand is very clear. I believe the authors, but this experiment should be repeated and hopefully a clearer result can be obtained. What also doesn't make sense is that in S5B there are two bands one that is found in every lane and then a unique shifted band near the top of the blot in the presence of the antibody. In Fig 5C the same band is observed in every well although it does appear darker in the lane where the antibody was labeled. As an alternative to repeating the blot, the authors could quantify the intensity of this band versus the same position in the control lanes to convince us that there is enrichment in their replicates.

3. Regarding the CRISPR experiment I question the value of using CRISPR/CAS9 over homologous recombination. I am under the impression that CRISPR/CAS9 is not actually providing any benefit and that the authors are observing homologous recombination. After transfection we expect the provided repair template to enter one of the two nuclei based on Poxleitner et al 2008 and Carpenter et al 2012. Therefore, a single transfection should be able to edit two copies of the genome. It is not expected to generate a true knockout due to the two nuclei challenge. Here however, PCR (8B) and western blotting (8E) both point toward just a single allele being disrupted since genomic DNA and protein levels are pretty close to the 25% reduction level of losing a single allele. The authors mention the use of NHEJ inhibiting drugs which seem to be doing nothing since Ebnetter et al showed that homologous recombination can be used to knockout genes in *Giardia*. I wonder why the authors bother to invoke the use of CRISPR since there is no perceivable benefit? Note that I do not dispute their mutant has a phenotype. My concern is that their pseudo use of CRISPR/CAS9 will confuse readers.

Minor Points

1. Fig S1 is corrupted or there was an error in its construction. The region around T328 is not shown and neither is the red arrow that is mentioned by the legend. Please fix this.

2. Sentence on Line 157 is confusing. It reads as if it is interesting that ESVs contain CWP1. To me what is interesting is that the localization of TOP3 β is more dispersed in this encysting cell.

3. Sentence on 342 is awkward please re-write

4. On 411 change catalytic to catalytically

5. I would change the last sentence of the conclusion. I don't believe this study told us much about the evolution of TOP3 β and whether TOP3 β is really a viable drug target. The function of the gene wasn't shown to be essential since only a single allele of TOP3 β was deleted according to the PCR in Fig 8B which roughly corresponds to the 30% reduction in TOP3 β protein levels. My recommendation: "Our study provides evidence for the important role of TOP3 β in the differentiation of *Giardia* trophozoites into cysts, leading to greater understanding of the mechanism regulating cell differentiation and parasite transmission."

Referee: 2

Comments to the Author(s)

The authors have submitted a detailed and interesting study of the effect of DNA topoisomerase 3b (type 1) on encystation in *Giardia*. The title focuses on the effect of cyst generation related to cwp expression. However, I'm not convinced that there is enough causal information to warrant the use of "by inducing" in the title.

They have used transfection studies to study the effect of upregulation of TOP3b on expression of the cwps and myb2 on encystation. They have also used CRISPR/Cas9 to reduce expression of top3b and shown decreases in the above. (Of note, knockout is impossible with *Giardia* because of its polyploidy, so these approaches are reasonable).

Overall, the scientific component of the work seems solid, but I do have some questions regarding the data to be addressed by the authors:

Thus, I will begin with the results section:

For figure 1 and other figures, the authors simply use $p < 0.05$, but unless < 0.001 , it would be helpful to see the numbers for CI and actual p value.

For Fig 4A, the difference between top3b positive and negative for linear DNA is not very convincing. Can the authors provide quantitative scan data from several runs?

Fig 5C, the mobility shifted lanes are not very convincing.

The bigger issues for correction relate to the writing:

1. The introduction has far too much of results in it. Almost 1/3 of the intro is results.
2. The biggest problem is the discussion. I felt like I was reading the results again when reading the discussion. There are two things that would greatly help. First, rather than reiterating results, explain the difficult parts and show how the results fit into other literature. Second, a figure with a model of what the authors think is happening would be very useful.
3. Although the writing is reasonably good, there are still occasional grammatical errors. If the journal has in-house copy editing, it should be ok.
4. A structured abstract would be nice if it fits with journal policy.

Author's Response to Decision Letter for (RSOB-19-0228.R0)

See Appendix A.

RSOB-19-0228.R1 (Revision)

Review form: Reviewer 1

Recommendation

Accept as is

Do you have any ethical concerns with this paper?

No

Comments to the Author

The authors have sufficiently addressed my concerns.

Decision letter (RSOB-19-0228.R1)

06-Jan-2020

Dear Dr SUN

We are pleased to inform you that your manuscript entitled "DNA topoisomerase III β promotes cyst generation by inducing cyst wall protein gene expression in *Giardia lamblia*" has been accepted by the Editor for publication in Open Biology.

If applicable, please find the referee comments below. No further changes are recommended.

Sincerely,
The Open Biology Team
mailto: openbiology@royalsociety.org

Reviewer(s)' Comments to Author:

Referee: 1

Comments to the Author(s)
The authors have sufficiently addressed my concerns.

Appendix A

Dear Journal editor,

Thank you very much for the critical review of our manuscript. We appreciate the comments and are submitting a manuscript revised according to the critiques and questions of the reviewers. The responses and the changes in the text are detailed below each critique. They are marked by //. The changed manuscript with highlight is shown below.

We hope that you find the revised manuscript suitable for publication in Open Biology.

Thank you very much,

Sincerely,

Chin-Hung Sun

12-Nov-2019

Dear Dr Sun,

We are writing to inform you that the Editor has reached a decision on your manuscript RSOB-19-0228 entitled "DNA topoisomerase III β promotes cyst generation by inducing cyst wall protein gene expression in *Giardia lamblia*", submitted to Open Biology.

As you will see from the reviewers' comments below, there are a number of criticisms that prevent us from accepting your manuscript at this stage. The reviewers suggest, however, that a revised version could be acceptable, if you are able to address their concerns. If you think that you can deal satisfactorily with the reviewer's suggestions, we would be pleased to consider a revised manuscript.

The revision will be re-reviewed, where possible, by the original referees. As such, please submit the revised version of your manuscript within four weeks. If you do not think you will be able to meet this date please let us know immediately.

When submitting your revised manuscript, please respond to the comments made by the referee(s) and upload a file "Response to Referees" in "Section 6 - File Upload". You can use this to document any changes you make to the original manuscript. In order to expedite the processing of the revised manuscript, please be as specific as possible in your response to the referee(s).

Please see our detailed instructions for revision requirements
<https://royalsociety.org/journals/authors/author-guidelines/>

Sincerely,

The Open Biology Team
mailto: openbiology@royalsociety.org

Reviewer(s)' Comments to Author(s):

Referee: 1
Comments to the Author(s)

Sun et al explore the role of TOP3 β in regulating Giardia differentiation. They demonstrate that TOP3 β localizes to the nucleus and binds to AT rich promoter sequences. They also show that over expression of this enzyme leads to up-regulation of CWP proteins and Myb2. Conversely

knockdown with CRISPR based allele deletion results in reduced CWP and Myb2 expression. Overall the results are clearly presented. I reviewed an earlier version of this paper and the authors addressed my major criticisms, so I recommend acceptance with some minor revisions I will outline below.

Major Points

1. The change in localization observed for TOP3 β in non-encysting cells (Fig 1) versus encysting cells in Sup Fig 3 is fascinating and this deserves more attention.

// The mentioned Fig. 1 is Fig. 2B. In Fig. 2B-F, both the Veg (vegetative) and Enc (encysting) stages are shown. In Fig. S3B (finally moved to Fig. 2C), we repeated the Enc (encysting) part with CWP1 costaining and confirmed the same cell with expression of both TOP3 β -HA and CWP1.

That TOP3 β localization changes from perinuclear to dispersed in the nucleus and is an important point that supports the authors assertion that TOP3 β is involved in regulating differentiation.

// The old Fig. S3B (finally moved to Fig. 2C) is a bit misleading, so we changed it to a new one. The localization of TOP3 β is still in the nuclear periphery. In Fig. 2B, The same “nuclear periphery” localization of cells in both the Veg (vegetative) and Enc (encysting) stages was also shown.

// The perinuclear localization of the Giardia TOP3 β may have its specific function. We wrote “Similarly, DNA topoisomerase I from yeast and human have been found to have a perinuclear distribution that may help function in DNA replication with perinuclear anchors of chromosomes (35).” (Line154)

This same re-localization is not observed in Fig 1 perhaps because the cell is not at the right stage of encystation or is not actually encysting (no CWP1 staining here).

// As we suggested above, there is no re-localization (we found the same “nuclear periphery” localization of TOP3 β in Fig. 2B (including Veg and Enc) and Fig. S3B (finally moved to Fig. 2C)). The encysting cell with CWP1 staining is shown in Fig. S3B (finally moved to Fig. 2C).

The cell shown in Fig S3 is sort of mid-encystation based on the CWP1 staining.

// We have changed the Fig S3B (finally moved to Fig. 2C). The encystation stage is based on the “positive” CWP1 staining as it is hard for us to classify the stage.

So I also wonder if the nuclear distribution changes with stage in the encystation response. This is potentially a really nice observation that if moved to Fig 1 will strengthen the paper because it shows TOP3 β is specifically changing localization in actively differentiating cells.

// We finally moved the Fig. S3B to Fig. 2C.

I would like the authors to include this result in Fig 1 and also quantify the number of cells that have dispersed nuclear versus perinuclear localization during encystation.

// We observed the only one kind of localization of TOP3 β : “nuclear periphery” in Fig. 2B (including Veg and Enc) and Fig. S3B (finally moved to Fig. 2C)), so we did not quantify it.

It is also interesting that the mutants are more clearly nuclear localized during encystation. Since there is no quantification of this I don't know if this is due to selected images or a real response to encystation stimulus.

// As we mentioned in paper (lines 159-165). TOP3 β m1 is also in nuclear periphery. TOP3 β m2 and m3 have less nuclear peripheral staining. We found consistent staining results for the same staining pattern, so we did not quantify it.

2. The super-shift assay in Fig 5C is not very convincing. A similar experiment in Fig S5B on the other hand is very clear. I believe the authors, but this experiment should be repeated and hopefully a clearer result can be obtained.

// We have changed the Fig. 5C with better image. This is the best gel from 3 same experiments. We have repeated the assays at least two times with similar results as shown in both Fig. 5C (in which double-stranded cwp1-45/-1 was used as the DNA probe) and Fig. S5B (in which single-stranded cwp3-30/+10 was used as the DNA probe). Since the double-stranded cwp1-45/-1 probe is different from the single-stranded cwp3-30/+10 probe in strand number, sequence, and

length, the bound forms for TOP3 β are also different. The former are more indistinct (Fig. 5C, arrowheads). The supershifts in both Figures are quite obvious as indicated by arrows.

What also doesn't make sense is that in S5B there are two bands one that is found in every lane and then a unique shifted band near the top of the blot in the presence of the antibody. In Fig 5C the same band is observed in every well although it does appear darker in the lane where the antibody was labeled. As an alternative to repeating the blot, the authors could quantify the intensity of this band versus the same position in the control lanes to convince us that there is enrichment in their replicates.

// In supershift assays, the anti-TOP3 β was added in the reaction and the size of supershift complex (anti-TOP3 β - TOP3 β -DNA probe) will be bigger than the bound form (TOP3 β - DNA probe) and may get stuck near the loading well (arrows in Fig. 5C and Fig. S5B). We quantified the region with arrows and showed the results in Fig. S5C. We added the text: "Both supershift results for the probes cwp1-45/-1 and cwp3 -30/+10F were significant as quantified in Fig. S5." (page 13)

2. Regarding the CRISPR experiment I question the value of using CRISPR/CAS9 over homologous recombination. I am under the impression that CRISPR/CAS9 is not actually providing any benefit and that the authors are observing homologous recombination.

// We hope our thorough analysis can provide a better method for the protozoan Giardia, so we must do a lot of work on the CRISPR/Cas9 system using mlf genes as a model target gene (Lin et al., 2019)(ref 11). Due to the tetraploid genome in two nuclei of Giardia, it could be hard to disrupt a gene completely in Giardia. We only generated knockdown but not knockout mutants. The potential of CRISPR/Cas9 system to complete knockout genes of interest in Giardia awaits further studies to explore. Using a strong promoter to drive the expression of cas9 gene could improve knockout efficiency.

// The good part for our CRISPR/Cas9 system is: Because of the integration of pac gene in genomic DNA, the knockdown effect may last very long time. We also show the similar knockdown effect after removal of puromycin for a month (Lin et al., 2019)(ref 11).

// A CRISPRi system for stable transcriptional repression of a target gene in Giardia also has been developed (McInally SG, Hagen KD, Nosala C, Williams J, Nguyen K, Booker J, et al. Robust

and stable transcriptional repression in Giardia using CRISPRi. Mol Biol Cell. 2018 Oct 31;mbcE18090605.). We added this ref in our CRISPR paper (Lin et al., 2019)(ref 11).

After transfection we expect the provided repair template to enter one of the two nuclei based on Poxleitner et al 2008 and Carpenter et al 2012. Therefore, a single transfection should be able to edit two copies of the genome. It is not expected to generate a true knockout due to the two nuclei challenge. Here however, PCR (8B) and western blotting (8E) both point toward just a single allele being disrupted since genomic DNA and protein levels are pretty close to the 25% reduction level of losing a single allele.

// The results from papers “Poxleitner et al 2008 and Carpenter et al 2012” are for analyzing genetic exchange occurs in cysts, not in trophozoites. For that study, they wanted to ensure the presence of only a single integrated construct in a trophozoite before encysting experiment. They tested clonal populations with PCR, and tested for the absence of episomes by PCR, and analyzed with fluorescence in situ hybridization (FISH). Therefore, they found a single nuclei staining.

// In our TOP3 β paper, our aim is to disrupt the top3 β gene for functional analysis. We did not make clonal populations. We did not ensure the presence of only a single integrated construct in a trophozoite. We did not ensure the absence of episomes. Therefore, there could be some mixtures of successful and unsuccessful cells. Interestingly we still got 23% disruption efficiency.

The authors mention the use of NHEJ inhibiting drugs which seem to be doing nothing since Ebnetter et al showed that homologous recombination can be used to knockout genes in Giardia.

// Cre-lox method in Ebnetter et al 2016 is also a very good method. We don't have a chance to work on Cre-lox. We wrote in our CRISPR paper (Lin et al., 2019)(ref 11): “A Cre/loxP system allows persistence of gene disruption in the absence of drug selection in Giardia (Wampfler et al., 2014; Ebnetter et al., 2016). It is interesting to compare CRISPR/Cas9 and Cre-lox systems in Giardia.”

I wonder why the authors bother to invoke the use of CRISPR since there is no perceivable benefit? Note that I do not dispute their mutant has a phenotype. My concern is that their pseudo use of CRISPR/CAS9 will confuse readers.

// As we wrote above, there is always a need for gene disruption techniques in Giardia. That is why CRISPRi has also been developed. For our lab, disruption of top3 β gene is the second successful example (the first one is mlf gene (Lin et al., 2019)(ref 11)). We wanted to check out how many genes can be successfully target-disrupted by CRISPR/Cas9.

Minor Points

1. Fig S1 is corrupted or there was an error in its construction. The region around T328 is not shown and neither is the red arrow that is mentioned by the legend. Please fix this.

// We have changed Fig. S1.

2. Sentence on Line 157 is confusing. It reads as if it is interesting that ESVs contain CWPI. To me what is interesting is that the localization of TOP3 β is more dispersed in this encysting cell.

// We tried to emphasize the role of TOP3 β by writing “Interestingly, the CWPI protein was stained in the ESVs of TOP3 β -HA positive stained cells (Fig. 2C), suggesting that TOP3 β may function in inducing the ESV and thereby in inducing cyst formation.”

//As we mentioned above, the old Fig. S3B (finally moved to Fig. 2C) is a bit misleading, so we changed it to a new one. The localization of TOP3 β is still in the nuclear periphery. In Fig. 2B, The same “nuclear periphery” localization of cells in both the Veg (vegetative) and Enc (encysting) stages was also observed.

3. Sentence on 342 is awkward please re-write

// We have changed it. (line 342)

4. On 411 change catalytic to catalytically

// We have changed it. (line 411)

5. I would change the last sentence of the conclusion. I don't believe this study told us much about the evolution of TOP3 β and whether TOP3 β is really a viable drug target. The

function of the gene wasn't shown to be essential since only a single allele of TOP3 β was deleted according to the PCR in Fig 8B which roughly corresponds to the 30% reduction in TOPO3 β protein levels. My recommendation: "Our study provides evidence for the important role of TOP3 β in the differentiation of Giardia trophozoites into cysts, leading to greater understanding of the mechanism regulating cell differentiation and parasite transmission."

// Many thanks for the suggestion. We have changed it. (lane 462)

Referee: 2

Comments to the Author(s)

The authors have submitted a detailed and interesting study of the effect of DNA topoisomerase 3b (type 1) on encystation in Giardia. The title focuses on the effect of cyst generation related to cwv expression. However, I'm not convinced that there is enough causal information to warrant the use of "by inducing" in the title.

They have used transfection studies to study the effect of upregulation of TOP3b on expression of the cwps and myb2 on encystation. They have also used CRISPR/Cas9 to reduce expression of top3b and shown decreases in the above. (Of note, knockout is impossible with Giardia because of its polyploidy, so these approaches are reasonable).

Overall, the scientific component of the work seems solid, but I do have some questions regarding the data to be addressed by the authors:

Thus, I will begin with the results section:

For figure 1 and other figures, the authors simply use $p < 0.05$, but unless < 0.001 , it would be helpful to see the numbers for CI and actual p value.

// We have added p value and CI (error bars) in every figure.

For Fig 4A, the difference between top3b positive and negative for linear DNA is not very

convincing. Can the authors provide quantitative scan data from several runs?

// We have added quantitation data for the linear DNA bands. (Fig. 4A)

Fig 5C, the mobility shifted lanes are not very convincing.

// We have changed the Fig. 5C with better image. This is the best gel from 3 same experiments.

The bigger issues for correction relate to the writing:

1. The introduction has far too much of results in it. Almost 1/3 of the intro is results.

// We have cut it down. (page 5)

2. The biggest problem is the discussion. I felt like I was reading the results again when reading the discussion. There are two things that would greatly help. First, rather than reiterating results, explain the difficult parts and show how the results fit into other literature. Second, a figure with a model of what the authors think is happening would be very useful.

// We have cut it down. (Discussion part)

// We have added a model in Fig. 9. (for text, please see page 17)

3. Although the writing is reasonably good, there are still occasional grammatical errors. If the journal has in-house copy editing, it should be ok.

// We have checked the correctness of the grammar.

4. A structured abstract would be nice if it fits with journal policy.

// We have checked the Journal policy. It does not require structured abstract, so we did not change the abstract.

[revised manuscript text omitted]

During *Giardia* encystation, a trophozoite with 2 nuclei (4N) may differentiate into a cyst
with 4 nuclei (16N) by DNA replication and homologous recombination may occur in the cyst

nuclei (1,27). Because type I topoisomerases play a critical role in cell differentiation (20,21,22),
we asked whether type I topoisomerases could be important for *Giardia* encystation. In our
previous study, we found that a *Giardia* type II topoisomerase (TOPO II) is an important factor
involved in inducing encystation and the TOPO II inhibitor, etoposide, can inhibit *Giardia*
growth and encystation (28). In this study, we further tried to understand the role of a type IA
topoisomerase, TOP3 β . We found that ~~the expression of the *Giardia* TOP3 β protein increased~~
~~during *Giardia* encystation. TOP3 β had DNA-binding and cleavage activity, and overexpression~~
~~of TOP3 β increased expression of *cwp1-3* and *myb2* and cyst formation. We used an approach~~
~~similar to chromatin immunoprecipitation (ChIP) assays, norfloxacin-mediated topoisomerase~~
~~immunoprecipitation assays (29) to confirm the association of TOP3 β with these gene promoters~~
~~*in vivo*. Oligonucleotide microarrays confirmed the up-regulation of the *cwp1-3* and *myb2* genes~~
~~and identified up-regulation of many genes in the TOP3 β -overexpressing cell line. Using~~
~~mutation analysis and CRISPR/Cas9 system, we found evidence of TOP3 β in inducing *Giardia*~~
~~encystation. We also tested the effect of a type IA topoisomerase inhibitor, norfloxacin, and~~
found that it inhibited *Giardia* growth and cyst formation, and increased the formation of
cleavage complex of TOP3 β and DNA. ~~Using mutation analysis and CRISPR/Cas9 system, we~~
~~found evidence of TOP3 β in inducing *Giardia* encystation.~~ Our results provide insights into the
role of TOP3 β in activation of *cwp* genes during *Giardia* encystation and into the effect of the
TOP3 β inhibitor, norfloxacin, on *Giardia* cyst formation and growth.

**Results**

**Identification and characterization of *top3β* gene.** Four putative homologues for
topoisomerases have been found in the *G. lamblia* genome database (28). One is Topo II
topoisomerase (open reading frame, orf, 16975). Orfs 15190 and 7615 are annotated as
topoisomerase III, which belongs to type IA topoisomerases. Sequence analysis suggests that orfs
15190 and 7615 are similar to human TOP3β and TOP3α, respectively (see below). The last
putative topoisomerase homologue is annotated as spo11 Type II DNA topoisomerases VI
subunit A. We focused on understanding the role of orf 15190 (topoisomerase IIIβ, TOP3β) in
*Giardia*. The deduced *Giardia* TOP3β protein contains 973 amino acids with a predicted
molecular mass of ~ 107.06 kDa and a pI of 8.40. It has a Toprim domain (residues 2 to 145) and
a DNA topoisomerase domain (residues 159 to 610) as predicted by Pfam (Fig.

[revised manuscript text omitted]

(~0.8-fold) (Fig. 3E). We found that 93 and 40 genes were significantly up-regulated (≥ 2 -fold)
and down-regulated ($\leq 1/2$) ($p < 0.05$) in the TOP3 β -overexpressing cell line relative to the vector
control, respectively (Table S1). The *top3 β* mRNA expression increased by ~2.1-fold ($p < 0.05$) in
the TOP3 β -overexpressing cell line (Fig. 3E).

**TOP3 β has DNA cleavage activity.** The type I topoisomerases have ability to bind to and
cleave single-stranded DNA (12, 13, 38). *Drosophila* TOP3 β cleaves DNA by forming a covalent
topoisomerase-DNA complex (39). To test DNA cleavage activity of TOP3 β , we expressed
TOP3 β in *E. coli* and purified it to >95% homogeneity. We performed DNA cleavage assays with
purified recombinant TOP3 β and pBluescript SK(+) plasmid. As shown in Fig. 4A, TOP3 β has
DNA cleavage activity.

Norfloxacin, a type II topoisomerase inhibitor, also inhibits *E. coli* topoisomerase I (type IA)
at higher concentrations, resulting in anti-bacteria activity (25). Norfloxacin can inhibit
topoisomerases by stabilizing covalent topoisomerase-DNA cleavage complexes (26). To
understand whether norfloxacin can inhibit *Giardia* TOP3 β DNA cleavage activity, we also
performed DNA cleavage assays with norfloxacin. As shown in Fig. 4B, the addition of
norfloxacin increased the amount of linear DNA, suggesting that norfloxacin can stabilize the
TOP3 β -DNA cleavage complex. We also tried to understand whether the products are from the
covalent TOP3 β -DNA cleavage complex. In a normal condition of the cleavage assay, proteinase
K was included to stop the reaction by removing TOP3 β from the cleavage complex (Fig. 4C).
When proteinase K was not included, the TOP3 β -DNA cleavage complex can not enter the gel
(Fig. 4C), suggesting that TOP3 β can form a cleavage complex with DNA. The results indicate

that TOP3 β may function as a topoisomerase in *Giardia*.

**Norfloxacin has anti-*Giardia* effect.** Norfloxacin is an inhibitor of type IA and type II
topoisomerases with anti-bacteria activity (25). We found that norfloxacin increased DNA
cleavage activity of TOP3 β , indicating that norfloxacin can trap the cleavage complex of TOP3 β
(Fig. 4B). We also found that treatment with norfloxacin significantly reduced *Giardia*
trophozoites growth (Fig. 4D). The half-maximal inhibitory concentration (IC₅₀) of norfloxacin

[revised manuscript text omitted]

*cwp1-3* and *myb2* mRNA and cyst formation (Fig. 3A, Fig. 3B, Fig. 3C, Fig. 3D). Results from
the CRISPR/Cas9 system suggest a decrease in expression of CWP1, *cwp1-3* and *myb2*, and cyst
formation by targeted disruption of the *top3 β* gene (Fig. 8E, Fig. 8F, Fig. 8G). Furthermore, the
addition of an inhibitor of type IA topoisomerases, norfloxacin, inhibited cell growth and cyst
formation (Fig. 4D, Fig. S3). The results suggest a positive role of TOP3 β in inducing *cwp1-3*
and *myb2* gene expression and *Giardia* encystation.

Many *Giardia* gene promoters have the AT-rich initiator elements responsible for promoter
activity and transcription start site selection (40,57,58). We have identified several transcription
factors involved in the transactivation of the *cwp* genes, and they can bind to the AT-rich
elements or the proximal upstream regions of the *cwp* promoters (7, 8,9,10,59,60,61). ~~It has been~~
~~reported that type IA topoisomerases can bind to DNA and activate gene expression (31,32, 50).~~
~~In this study, we found that TOP3 β can also bind to the AT-rich elements of the *cwp* promoters *in*~~
~~*vitro* (Fig. 6). Norfloxacin-mediated topoisomerase immunoprecipitation assays confirmed the~~

~~association of TOP3 β with its own promoter and the *cwp1-3* and *myb2* promoters but not with~~
~~the U6 promoter *in vivo* (Fig. 7A, Fig. 7B).~~ChIP assays confirmed the binding of
encystation-induced transcription factors E2F1 and MYB2 to the *cwp* and *myb2* gene promoters
previously (7,10). E2F1 and MYB2 may interact together to activate expression of the *cwp* genes
(10). We also found that MYB2 is co-immunoprecipitated with TOP3 β (Fig. 7D). Treatment with
DNase did not prevent the immunoprecipitation of Myb2 with TOP3 β (Fig. 7D), suggesting that
the interaction depends on protein-protein interaction but not DNA. Far Western blot analysis, a
non-antibody method, was further used to confirm this interaction between the MYB2-N and
TOP3 β -HA to avoid the nonspecific problem (Fig. 7E). The MYB2-N can be phosphorylated by
CDK2, which is involved in inducing encystation (6). ~~The interaction of TOP3 β and MYB2, and~~
~~other transcription factors binding to the AT-rich elements or the proximal upstream regions (7),~~
~~may be required for promoter activity and accurate transcription start site selection.~~TOP3 β can
bind to AT-rich elements of both the constitutive *ran* gene and encystation-induced *cwp* genes
(Fig. 6). However, overexpressed TOP3 β can induce the CWP1 level but not the RAN level (Fig.
3A). This could be due to a lack of cooperation of the encystation-specific transcription factors to
transactivate the constitutive *ran* gene. Similar results were found in studies of other
transcription factors (7, 8,9,10,60,61). Interestingly, expression of all three TOP3 β mutants
(TOP3 β m1-3) have led to less expression levels of CWP1, cyst formation, and *cwp1-3* and *myb2*
mRNA relative to the wild-type TOP3 β (Fig. 3A, Fig. 3B, Fig. 3C, Fig. 3D). However, they it
still have led to more expression levels of CWP1, cyst formation, and *cwp1-3* and *myb2* mRNA
relative to the vector control (Fig. 3A, Fig. 3B, Fig. 3C, Fig. 3D). It is possible that the mutants
that were relatively overexpressed, may still interact with transcription factors, such as Myb2, to

activate expression of the *cwp* genes.

We also found that the *top3β* promoter contains the MYB2 binding sequences (Fig. S10)(61),
suggesting that *top3β* gene expression is up-regulated by MYB2 and that TOP3β might play a
positive role in *Giardia* encystation. Since overexpressed TOP3β increased the MYB2 level (Fig.
3A), there is a positive regulation cycle between TOP3β and MYB2. The human *top3α*
promoters also contain the binding sequence of YY1 and USF1 activators, which are important
for cell growth and differentiation (62). The induction ability of the overexpressed TOP3β in *cwp*
transcription was active in vegetative and encystation stages (Fig. 3 and Fig. S3). Similarly,
Myb2 and Pax1 transcription factors in expression system can also induce *cwp* transcription in
both stages (9,28). This suggests that the specific promoter or enhancer elements for *cwp*
transcription may be active in both stages.

~~The DNA topoisomerase domain (residues 159 to 610) of *Giardia* TOP3β is near the central-~~
~~region (Fig. 1A). A zinc ribbon domain is located in the C terminus (residues 645 to 973)(Fig-~~
~~S1)-~~ Studies suggest that the C-terminal zinc ribbon domains of *E. coli* topoisomerase I and
*Drosophila* topoisomerase IIIα are important for DNA binding (31, 32). The C-terminal zinc
ribbon domain of *E. coli* topoisomerase I interacts with RNA polymerase to help bring to the
transcription site for relaxation reaction (18). ~~We found that D~~deletion of the C-terminal 332
amino acids (residues 642-973) corresponding to the zinc ribbon domain of TOP3β (residues-
642-973) resulted in reduction in DNA-binding activity, but only slight decrease in cleavage
activity (TOP3βm2) (Fig. 5D, Fig. S1), suggesting that the zinc ribbon domain is important for
DNA-binding. Deletion of C-terminal 552 amino acids (residues 422-973) corresponding to the
zinc ribbon domain and a part of topoisomerase domain of TOP3β (residues 422-973) resulted in

reduction in the cleavage activity (TOP3 β m3) (Fig. 1A, Fig. 5B), indicating that the
topoisomerase domain is important for cleavage activity. We also found that a mutation of the
catalytically important Tyr 328 resulted in a decrease in cleavage activity (TOP3 β m1) (Fig. 1A,
Fig. 5B). Tyr327 of topoisomerase III β in *Leishmania donovani* is also catalytically important
(63). Interestingly, mutation of the Tyr 328 to Phe did not change its perinuclear localization in
both vegetative and encysting cells (TOP3 β m1) (Fig. 2C2D). Mutation of this important Tyr also
resulted in a significant decrease in the levels of CWP1, cyst formation, and *cwp1-3* and *myb2*
mRNA (Fig. 3A, Fig. 3B, Fig. 3C, Fig. 3D, Fig. 5B, Fig. 5D), suggesting a correlation of DNA
cleavage activity and *in vivo* function.

Typically, nuclear localization signal (NLS) is a region rich with basic amino acids. Two
putative NLS motifs were predicted in TOP3 β using the PSORT program (<http://www.psort.org/>),
including RKHR at 970, and RRAAQPKRHGPRGRKHR at 957. We also found that deletion of
the C-terminal 332 (residues 642-973) or 552 amino acids (residues 422-973) resulted in a
decrease but not complete loss of perinuclear localization (TOP3 β m2 or m3) (Fig. 2D2E, Fig.
2E2F), suggesting that the C-terminal zinc ribbon domain may play a partial role in the
perinuclear localization and that other NLS motifs may be present in TOP3 β . Deletion of the
C-terminal 332 amino acids (TOP3 β m2) resulted in a significant decrease in the levels of CWP1,
cyst formation, and *cwp1-3* and *myb2* mRNA (~~TOP3 β m2~~) (Fig. 3A, Fig. 3B, Fig. 3C, Fig. 3D),
but the effect is lower than the TOP3 β m1 and TOP3 β m3. As Shown in Fig. 2D2E, TOP3 β m2
still has some ability to localize to nuclear periphery. Interestingly, DNA cleavage activity was
less affected in this mutant (TOP3 β m2) (Fig. 5B), suggesting again- a correlation of DNA
cleavage activity and *in vivo* function. The results suggest that TOP3 β may enhance the

encystation-induced expression of *cwp1-3* and *myb2* through its cleavage activity.

~~*Drosophila* topoisomerase III β prefers to bind to AT-rich DNA sequences (39). We also found~~
~~that *Giardia* TOP3 β can bind to the AT-rich promoter elements of the *cwp* genes (Fig. 6), and~~
~~that it may bind to and up-regulate the *cwp* gene promoters to induce *Giardia* encystation.~~ We
found that 93 and 40 genes were significantly up-regulated and down-regulated in the
TOP3 β -overexpressing cell line relative to the vector control, respectively (Table S1). ~~In addition,~~
~~targeted disruption of the *top3 β* gene resulted in a decrease in expression of *cwp1-3* and *myb2*~~
~~and cyst formation using the CRISPR/Cas9 system (Fig. 8), suggesting TOP3 β may induce~~
~~*Giardia* encystation.~~ We also found that chemosensitivity of the TOP3 β td cell line significantly
increased by the addition of G418 (Fig. 8D), suggesting that TOP3 β may affect many genes
involved in cell growth to survive antibiotic stress.

Norfloxacin, which belongs to quinolones, is an inhibitor of the type II topoisomerases,
including topoisomerase II (DNA gyrase) and topoisomerase IV, ~~and (25). It~~ also inhibits type
IA topoisomerases at higher concentrations (25). Quinolones can stabilize the topoisomerase
II-DNA complex and prevent religation of DNA, resulting in anti-topoisomerase activity (23, 26,
64). A model suggests that quinolones can form complexes with DNA and topoisomerase IV and
create barriers to DNA replication (65). We found that the addition of norfloxacin increased the
cleavage complex formation of *Giardia* TOP3 β (Fig. 4B), and ~~that the addition of norfloxacin~~
significantly decreased cell growth and cyst formation (Fig. 4D, Fig. S3). The IC₅₀ of
norfloxacin is 497 μ M, which is similar to the IC₅₀ for mammalian cells (470 μ M for CT-26 cells)
(66). It is much higher than the IC₅₀ of norfloxacin against *E. coli* (0.3 μ M) and the IC₅₀ for
inhibiting *E. coli* topoisomerase I activity (135 μ M) (25). The difference of effective norfloxacin

concentrations between *Giardia* and *E. coli* could be due to the variability of the overall
sequences of *Giardia* TOP3 β and *E. coli* topoisomerase I, ~~in which the region~~ is helpful for
designing therapeutic drugs. Our results suggest that norfloxacin is less effective than the
standard drug metronidazole, ~~which has been used often in the treatment of *Giardia* infection~~
with an IC50 of 2.1 μ M (67,68).

[revised manuscript text omitted]
- 2. Einarsson, E., Ma'ayeh, S., and Svärd, S. G. (2016) An up-date on *Giardia* and giardiasis. *Curr.*
*Opin. Microbiol.* 34, 47-52.
- 3. Hanevik, K., Wensaas, K. A., Rortveit, G., Eide, G. E., Mørch, K., and Langeland, N. (2014)
Irritable bowel syndrome and chronic fatigue 6 years after giardia infection: a controlled
prospective cohort study. *Clin. Infect. Dis.* 59, 1394-1400.
- 4. Halliez, M. C., and Buret, A. G. (2013) Extra-intestinal and long term consequences of *Giardia*
*duodenalis* infections. *World J. Gastroenterol.* 19, 8974-8985.
- 5. Morrison, H. G., McArthur, A. G., Gillin, F. D., Aley, S. B., Adam, R. D., Olsen, G. J., Best, A.
736 A., Cande, W. Z., Chen, F., Cipriano, M. J., Davids, B. J., Dawson, S. C., Elmendorf, H. G., Hehl,
737 A. B., Holder, M. E., Huse, S. M., Kim, U. U., Lasek-Nesselquist, E., Manning, G., Nigam, A.,
Nixon, J. E., Palm, D., Passamaneck, N. E., Prabhu, A., Reich, C. I., Reiner, D. S., Samuelson, J.,
Svard, S. G., and Sogin, M. L. (2007) Genomic minimalism in the early diverging intestinal
parasite *Giardia lamblia*. *Science* 317, 1921-1926.
- 6. Cho, C. C., Su, L. H., Huang, Y. C., Pan, Y. J., and Sun, C. H. (2012) Regulation of a Myb
transcription factor by cyclin dependent kinase 2 in *Giardia lamblia*. *J. Biol. Chem.* 287,
3733-3750.
- 7. Huang, Y. C., Su, L. H., Lee, G. A., Chiu, P. W., Cho, C. C., Wu, J. Y., and Sun, C. H. (2008)
Regulation of cyst wall protein promoters by Myb2 in *Giardia lamblia*. *J. Biol. Chem.* 283,
31021-31029.
- 8. Pan, Y. J., Cho, C. C., Kao, Y. Y., and Sun, C. H. (2009) A novel WRKY-like protein involved
in transcriptional activation of cyst wall protein genes in *Giardia lamblia*. *J. Biol. Chem.* 284,
17975-17988.
- 9. Wang, Y. T., Pan, Y. J., Cho, C. C., Lin, B. C., Su, L. H., Huang, Y. C., and Sun, C. H. (2010) A
novel pax-like protein involved in transcriptional activation of cyst wall protein genes in *Giardia*
*lamblia*. *J. Biol. Chem.* 285, 32213-32226.
- 10. Su, L. H., Pan, Y. J., Huang, Y. C., Cho, C. C., Chen, C. W., Huang, S. W., Chuang, S. F., and
Sun, C. H. (2011) A Novel E2F-like Protein Involved in Transcriptional Activation of Cyst Wall
Protein Genes in *Giardia lamblia*. *J. Biol. Chem.* 286, 34101-34120.
- 11. Lin, Z. Q., Gan, S. W., Tung, S. Y., Ho, C. C., Su, L. H., and Sun, C. H. (2019) Development
of CRISPR/Cas9-mediated gene disruption systems in *Giardia lamblia*. *PLoS ONE.* 14,
e0213594
- 12. Champoux, J. J. (2001) DNA topoisomerases: structure, function, and mechanism. *Annu. Rev.*

Biochem. 70, 369-413.
13. Vos, S. M., Tretter, E. M., Schmidt, B. H., and Berger, J. M. (2011) All tangled up: how cells
direct, manage and exploit topoisomerase function. *Nat. Rev. Mol. Cell Biol.* 12, 827-841
14. Heng, X., Le, W. D. (2010) The function of DNA topoisomerase II β in neuronal development.
*Neurosci. Bull.* 26, 411-416.
15. Kawamura, R., Pope, L. H., Christensen, M. O., Sun, M., Terekhova, K., Boege, F., Mielke,
C., Andersen, A. H., and Marko, J. F. (2010) Mitotic chromosomes are constrained by
topoisomerase II-sensitive DNA entanglements. *J. Cell Biol.* 188, 653-663.
16. Goulaouic, H., Roulon, T., Flamand, O., Grondard, L., Lavelle, F., and Riou, J. F. (1999)
Purification and characterization of human DNA topoisomerase III α . *Nucleic Acids Res.* 27,
2443-2450.
17. Aravind, L., Leipe, D. D., and Koonin, E. V. (1998) Toprim--a conserved catalytic domain in
type IA and II topoisomerases, DnaG-type primases, OLD family nucleases and RecR proteins.
*Nucleic acids Res.* 26, 4205-4213.
18. Cheng, B., Zhu, C. X., Ji, C., Ahumada, A., and Tse-Dinh, Y. C. (2003) Direct interaction
between *Escherichia coli* RNA polymerase and the zinc ribbon domains of DNA topoisomerase I.
*J. Biol. Chem.* 278, 30705-30710.
19. Wallis, J. W., Chrebet, G., Brodsky, G., Rolfe, M., and Rothstein, R. (1989) A
hyper-recombination mutation in *S. cerevisiae* identifies a novel eukaryotic topoisomerase. *Cell*
58, 409-419
20. Kwan, K. Y., and Wang, J. C. (2001) Mice lacking DNA topoisomerase III β develop to
maturity but show a reduced mean lifespan. *Proc. Natl. Acad. Sci. U. S. A.* 98, 5717-5721.
21. Kwan, K. Y., Moens, P. B., and Wang, J. C. (2003) Infertility and aneuploidy in mice lacking
a type IA DNA topoisomerase III β . *Proc. Natl. Acad. Sci. U. S. A.* 100, 2526-2531
22. Mönnich, M., Hess, I., Wiest, W., Bachrati, C., Hickson, I. D., Schorpp, M., and Boehm, T.
(2010) Developing T lymphocytes are uniquely sensitive to a lack of topoisomerase III α .
*Eur. J. Immunol.* 40, 2379-2384.
23. Pommier, Y., Leo, E., Zhang, H., and Marchand, C. (2010) DNA topoisomerases and their
poisoning by anticancer and antibacterial drugs. *Chem. Biol.* 17, 421-433.
24. Delgado, J. L., Hsieh, C. M., Chan, N. L., and Hiasa, H. (2018) Topoisomerases as anticancer
targets. *Biochem. J.* 475, 373-398.
25. Tabary, X., Moreau, N., Dureuil, C., and Le Goffic, F. (1987) Effect of DNA gyrase inhibitors
pefloxacin, five other quinolones, novobiocin, and clorobiocin on *Escherichia coli* topoisomerase
I. *Antimicrob. Agents Chemother.* 31, 1925-1928

26. Pohlhaus, J. R., and Kreuzer, K. N. (2005) Norfloxacin-induced DNA gyrase cleavage
complexes block *Escherichia coli* replication forks, causing double-stranded breaks in vivo. *Mol.*
*Microbiol.* 56, 1416–1429.

27. Carpenter, M. L., Assaf, Z. J., Gourguechon, S., and Cande, W. Z. (2012) Nuclear inheritance
and genetic exchange without meiosis in the binucleate parasite *Giardia intestinalis*. *J. Cell Sci.*
125, 2523-2532.

28. Lin, B. C., Su, L. H., Weng, S. C., Pan, Y. J., Chan, N. L., Li, T. K., Wang, H. C., and Sun, C.
H. (2013) DNA topoisomerase II is involved in regulation of cyst wall protein genes and
differentiation in *Giardia lamblia*. *PLoS Negl. Trop. Dis.* 7, e2218.

29. El Sayyed, H., and Espéli, O. (2018) Mapping *E. coli* Topoisomerase IV Binding and
Activity Sites. *Methods Mol. Biol.* 1703, 87-94.

30. Finn, R. D., Tate, J., Mistry, J., Coghill, P. C., Sammut, J. S., Hotz, H. R., Ceric, G., Forslund,
806 K., Eddy, S. R., Sonnhammer, E. L., and Bateman, A. (2008) The Pfam protein families database.
*Nucleic Acids Res.* 36, D281-D288

31. Zhang, H. L., Malpure, S., Li, Z., Hiasa, H., and DiGate, R. J. (1996) The role of the
carboxyl-terminal amino acid residues in *Escherichia coli* DNA topoisomerase III-mediated
catalysis. *J. Biol. Chem.* 271, 9039-9045.

32. Chen, S. H., Wu, C. H., Plank, J. L., and Hsieh, T. S. (2012) Essential functions of C
terminus of *Drosophila* Topoisomerase III α in double holliday junction dissolution. *J. Biol.*
*Chem.* 287, 19346-19353.

33. Lynn, R. M., Bjornsti, M. A., Caron, P. R., and Wang, J. C. (1989). Peptide sequencing and
site-directed mutagenesis identify tyrosine-727 as the active site tyrosine of *Saccharomyces*
*cerevisiae* DNA topoisomerase I. *Proc. Natl. Acad. Sci. U. S. A.* 86, 3559-3563.

34. Mondragón, A., and DiGate, R. (1999) The structure of *Escherichia coli* DNA topoisomerase
III. *Structure* 7, 1373-1383.

35. Hann, C., Evans, D. L., Fertala, J., Benedetti, P., Bjornsti, M. A., and Hall, D. J. (1998)
Increased camptothecin toxicity induced in mammalian cells expressing *Saccharomyces*
*cerevisiae* DNA topoisomerase I. *J. Biol. Chem.* 273, 8425-8433.

36. Singer, S. M., Yee, J., and Nash, T. E. (1998) Episomal and integrated maintenance of foreign
DNA in *Giardia lamblia*. *Mol. Biochem. Parasitol.* 92, 59-69.

37. Su, L. H., Lee, G. A., Huang, Y. C., Chen, Y. H., and Sun, C. H. (2007) Neomycin and
puromycin affect gene expression in *Giardia lamblia* stable transfection. *Mol. Biochem. Parasitol.*
156, 124-135.

38. Forterre, P., Gribaldo, S., Gabelle, D., and Serre, M. C. (2007) Origin and evolution of DNA

topoisomerases. *Biochimie*. 89, 427-446.

39. Wilson-Sali, T., and Hsieh, T. S. (2002) Generation of Double-stranded Breaks in
Hypernegatively Supercoiled DNA by *Drosophila* Topoisomerase III β , a Type IA Enzyme. *The J.*
*Biol. Chem.* 277, 26865-26871.

40. Sun, C. H., and Tai, J. H. (1999) Identification and characterization of a ran gene promoter in
the protozoan pathogen *Giardia lamblia*. *J. Biol. Chem.* 274, 19699-19706.

41. Gao, K., Chi, Y., Zhang, X., Zhang, H., Li, G., Sun, W., Takeda, M., and Yao, J. (2015) A
novel TXNIP-based mechanism for Cx43-mediated regulation of oxidative drug injury. *J. Cell*
*Mol. Med.* 19, 2469-2480.

42. Wang, J. C. (2002) Cellular roles of DNA topoisomerases: a molecular perspective. *Nat. Rev.*
*Mol. Cell. Biol.* 3, 430-440.

43. Suski, C., and Marians, K. J. (2008) Resolution of converging replication forks by RecQ and
topoisomerase III. *Mol. cell* 30, 779-789.

44. Viard, T., and de la Tour, C. B. (2007) Type IA topoisomerases: a simple puzzle? *Biochimie*
89, 456-467.

45. Baranello, L., Wojtowicz, D., Cui, K., Devaiah, B. N., Chung, H. J., Chan-Salis, K. Y., Guha,
R., Wilson, K., Zhang, X., Zhang, H., Piotrowski, J., Thomas, C. J., Singer, D. S., Pugh, B. F.,
Pommier, Y., Przytycka, T. M., Kouzine, F., Lewis, B. A., Zhao, K., and Levens, D. (2016) RNA
Polymerase II Regulates Topoisomerase I Activity to Favor Efficient Transcription. *Cell* 165,
357-371.

46. Harmon, F. G., DiGate, R. J., and Kowalczykowski, S. C. (1999) RecQ helicase and
topoisomerase III comprise a novel DNA strand passage function: a conserved mechanism for
control of DNA recombination. *Mol. Cell* 3, 611-620.

47. Ira, G., Malkova, A., Liberi, G., Foiani, M., and Haber, J. E. (2003) Srs2 and Sgs1-Top3
suppress crossovers during double-strand break repair in yeast. *Cell* 115, 401-411.

48. Wu, L., and Hickson, I. D. (2003) The Bloom's syndrome helicase suppresses crossing-over
during homologous recombination. *Nature* 426, 870-874.

49. Juge, F., Fernando, C., Fic, W., and Tazi, J. (2010) The SR protein B52/SRp55 is required for
DNA topoisomerase I recruitment to chromatin, mRNA release and transcription shutdown.
*PLoS Genet.* 6, e1001124.

50. Durand-Dubief, M., Persson, J., Norman, U., Hartsuiker, E., and Ekwall, K. (2010)
Topoisomerase I regulates open chromatin and controls gene expression in vivo. *EMBO J.* 29,
2126-2134.

51. Ahmed, W., Sala, C., Hegde, S. R., Jha, R. K., Cole, S. T., and Nagaraja, V. (2017)

Transcription facilitated genome-wide recruitment of topoisomerase I and DNA gyrase. *PLoS*
*Genet.* 13, e1006754.

52. Yang, Y., McBride, K. M., Hensley, S., Lu, Y., Chedin, F., and Bedford, M. T. (2014)
Arginine methylation facilitates the recruitment of TOP3B to chromatin to prevent R loop
accumulation. *Mol. Cell* 53, 484-497.

53. Huang, L., Wang, Z., Narayanan, N., and Yang, Y. (2018) Arginine methylation of the
C-terminus RGG motif promotes TOP3B topoisomerase activity and stress granule localization.
*Nucleic Acids Res.* 46, 3061-3074.

54. Lujan, H. D., Mowatt, M. R., Conrad, J. T., Bowers, B., and Nash, T. E. (1995) Identification
of a novel *Giardia lamblia* cyst wall protein with leucine-rich repeats. Implications for secretory
granule formation and protein assembly into the cyst wall. *J. Biol. Chem.* 270, 29307-29313.

55. Mowatt, M. R., Lujan, H. D., Cotton, D. B., Bower, B., Yee, J., Nash, T. E., and Stibbs, H. H.
(1995) Developmentally regulated expression of a *Giardia lamblia* cyst wall protein gene. *Mol*
*Microb.* 15, 955-963.

56. Sun, C. H., McCaffery, J. M., Reiner, D. S., and Gillin, F. D. (2003) Mining the *Giardia*
*lamblia* genome for new cyst wall proteins. *J. Biol. Chem.* 278, 21701-21708.

57. Yee, J., Mowatt, M. R., Dennis, P. P., Nash, T. E. (2000) Transcriptional analysis of the
glutamate dehydrogenase gene in the primitive eukaryote, *Giardia lamblia*. Identification of a
primordial gene promoter. *J. Biol. Chem.* 275, 11432-11439.

58. Elmendorf, H. G., Singer, S. M., Pierce, J., Cowan, J., and Nash, T. E. (2001) Initiator and
upstream elements in the alpha2-tubulin promoter of *Giardia lamblia*. *Mol. Biochem. Parasitol.*
113, 157-169.

59. Wang, C. H., Su, L. H., and Sun, C. H. (2007) A novel ARID/Bright-like protein involved in
transcriptional activation of cyst wall protein 1 gene in *Giardia lamblia*. *J. Biol. Chem.* 282,
8905-8914.

60. Chuang, S. F., Su, L. H., Cho, C. C., Pan, Y. J., and Sun, C. H. (2012) Functional
Redundancy of Two Pax-like Proteins in Transcriptional Activation of Cyst Wall Protein Genes
in *Giardia lamblia*. *PLoS ONE* 7, e30614.

61. Sun, C. H., Palm, D., McArthur, A. G., Svård, S. G., and Gillin, F. D. (2002) A novel
Myb-related protein involved in transcriptional activation of encystation genes in *Giardia lamblia*.
*Mol. Microbiol.* 46, 971-984.

62. Park, E. J., Han, S. Y., Chung, I. K. (2001) Regulation of mouse DNA topoisomerase IIIalpha
gene expression by YY1 and USF transcription factors. *Biochem. Biophys. Res. Commun.* 283,
384-391.

63. Banerjee, B., Sen, N., and Majumder, H.K. (2011) Identification of a functional type IA
topoisomerase, LdTopIIIb, from kinetoplastid parasite *Leishmania donovani*. *Enzyme research*
2011, 230542.

64. Barrett, J. F., Gootz, T. D., McGuirk, P. R., Farrell, C. A., and Sokolowski, S. A. (1989) Use
of in vitro topoisomerase II assays for studying quinolone antibacterial agents. *Antimicrob.*
*Agents Chemother.* 33, 1697-1703.

65. Hooper, D. C. (2001) Mechanisms of action of antimicrobials: focus on fluoroquinolones.
*Clin. Infect. Dis.* 32 Suppl 1, S9-S15.

66. Bykowska, A., Starosta, R., Komarnicka, U. K., Ciunik, Z., Kyzioł, A., Guz-Regner, K.,
Bugla-Płoskońskac, G., and Jeżowska-Bojczuk, M. (2014) Phosphine derivatives of
ciprofloxacin and norfloxacin, a new class of potential therapeutic agents. *New J. Chem.* 38,
1062–1071.

67. Upcroft, J. A. (2011) Pyruvate:ferredoxin oxidoreductase and thioredoxin reductase are
involved in 5-nitroimidazole activation while flavin metabolism is linked to 5-nitroimidazole
resistance in *Giardia lamblia*. *J. Antimicrob. Chemother.* 66, 1756-1765.

68. Bell, C. A., Cory, M., Fairley, T. A., Hall, J. E., and Tidwell, R. R. (1991) Structure-activity
relationships of pentamidine analogs against *Giardia lamblia* and correlation of anti-giardial
activity with DNA-binding affinity. *Antimicrob. Agents Chemother.* 35, 1099-1107.

69. Keister, D. B. (1983) Axenic culture of *Giardia lamblia* in TYI-S-33 medium supplemented
with bile. *Trans. R. Soc. Trop. Med. Hyg.* 77, 487-488.

70. Gillin, F. D., Boucher, S. E., Rossi, S. S., and Reiner, D. S. (1996) *Giardia lamblia*: the roles
of bile, lactic acid, and pH in the completion of the life cycle in vitro. *Exp. Parasitol.* 69,
164-174.

71. McArthur, A. G., Morrison, H. G., Nixon, J. E., Passamaneck, N. Q., Kim, U., Hinkle, G.,
Crocker, M. K., Holder, M. E., Farr, R. Reich, C. I., Olsen, G. E., Aley, S. B., Adam, R. D., Gillin,
F. D., and Sogin, M. L. (2000) The *Giardia* genome project database. *FEMS Microbiol. Lett.* 189,
271-273.

72. Chen, Y. H., Su, L. H., and Sun, C. H. (2008) Incomplete nonsense-mediated mRNA decay in
*Giardia lamblia*. *Int. J. Parasitol.* 38, 1305-1317.

73. Sun, C. H., Chou, C. F., and Tai, J. H. (1998) Stable DNA transfection of the primitive
protozoan pathogen *Giardia lamblia*. *Mol. Biochem. Parasitol.* 92, 123-132.

74. Knodler, L. A., Svard, S. G., Silberman, J. D., Davids, B. J., and Gillin, F. D. (1999)

[revised manuscript text omitted]

格式化: 缩排: 第一行: 1 字元

1177
1178
1179